# Reconstructed Ir–O–Mo species with strong Brønsted acidity for acidic water oxidation

Shiyi Chen[1,2,5], Shishi Zhang[1,2,5], Lei Guo[1,2], Lun Pan ●[1,2,3], Chengxiang Shi[1,2,3], Xiangwen Zhang[1,2,3], Zhen-Feng Huang ●[1,2,3] ✉, Guidong Yang ●[4] ✉ & Ji-Jun Zou ●[1,2,3] ✉

Surface reconstruction generates real active species in electrochemical conditions; rational regulating reconstruction in a targeted manner is the key for constructing highly active catalyst. Herein, we use the high-valence Mo modulated orthorhombic $Pr_3Ir_{1-x}Mo_xO_7$ as model to activate lattice oxygen and cations, achieving directional and accelerated surface reconstruction to produce self-terminated Ir–$O_{bri}$–Mo ($O_{bri}$ represents the bridge oxygen) active species that is highly active for acidic water oxidation. The doped Mo not only contributes to accelerated surface reconstruction due to optimized Ir–O covalency and more prone dissolution of Pr, but also affords the improved durability resulted from Mo-buffered charge compensation, thereby preventing fierce Ir dissolution and excessive lattice oxygen loss. As such, Ir–$O_{bri}$–Mo species could be directionally generated, in which the strong Brønsted acidity of $O_{bri}$ induced by remaining Mo assists with the facilitated deprotonation of oxo intermediates, following bridging-oxygen-assisted deprotonation pathway. Consequently, the optimal catalyst exhibits the best activity with an overpotential of 259 mV to reach 10 mA $cm_{geo}^{-2}$, 50 mV lower than undoped counterpart, and shows improved stability for over 200 h. This work provides a strategy of directional surface reconstruction to constructing strong Brønsted acid sites in $IrO_x$ species, demonstrating the perspective of targeted electrocatalyst fabrication under in situ realistic reaction conditions.

Electrocatalytic water splitting driven by renewable power sources provides an attractive strategy for green hydrogen production, contributing to a fully decarbonized society. Proton exchange membrane (PEM) water electrolyzers have merits of compact structure, high current density and energy efficiency, high operating pressure, and high-purity hydrogen[1–3]. However, the sluggish anodic oxygen evolution reaction (OER) in acidic medium is especially restricted to the high cost and scarcity of Ir-based oxides, impeding large-scale deployment. Consequently, it is critical to develop acidic OER electrocatalysts possessing both reduced iridium consumption and excellent electrocatalytic performance.

Improving the dispersion and intrinsic activity of active $IrO_x$ species are general strategies for constructing highly efficient electrocatalyst applied in acidic OER[2,4–6]. Highly dispersed Ir–$O^{(II-δ)-}$–Ir active species can be generated through electrochemical surface reconstruction of iridates like perovskite, pyrochlore and weberite structures[7–12]. Previous studies demonstrate that $IrO_6$ octahedra configurations of iridates govern the OER activity and longevity to a large

[1]Key Laboratory for Green Chemical Technology of the Ministry of Education, School of Chemical Engineering and Technology, Tianjin University, 300072 Tianjin, China. [2]Collaborative Innovative Centre of Chemical Science and Engineering (Tianjin), 300072 Tianjin, China. [3]Haihe Laboratory of Sustainable Chemical Transformations, 300192 Tianjin, China. [4]XJTU-Oxford International Joint Laboratory for Catalysis, School of Chemical Engineering and Technology, Xi'an Jiaotong University, Xi'an, Shaanxi, China. [5]These authors contributed equally: Shiyi Chen, Shishi Zhang. ✉e-mail: zfhuang@tju.edu.cn; guidongyang@mail.xjtu.edu.cn; jj_zou@tju.edu.cn

extent[10,13]. Iridates with weakly corner-shared $IrO_6$ configuration exhibit high initial lattice oxygen reactivity whereas dramatically deteriorate soon because of its kinetically sluggish regeneration, leading to drastic cations leach-out and continuous Ir loss[9,10]. Weberite type $Ln_3IrO_7$, featured with corner-linked $IrO_6$ octahedra lying along $c$-axis with shared apical $O_{(3)}$ and layered structure along $a$-axis, provides prerequisites to activate lattice oxygen redox under electrochemical conditions due to the large O $2p$ contribution around Fermi level (Fig. 1a, d and Supplementary Note 1)[11,14–16]. Charge compensation by valence-variable metals to fierce Ln cation leaching can effectively avoid excessive loss of lattice oxygen and further Ir dissolution. In addition, the mass content of iridium in $Ln_3IrO_7$ compounds (26.4% in $Pr_3IrO_7$) is obviously lower than that in $IrO_2$ (85.7%), perovskite and pyrochlore structures (58.6% in $SrIrO_3$ and 49.4% in $Pr_2Ir_2O_7$). Consequently, if properly tuned, directional surface reconstruction with improved stability and mass activity (normalized to Ir mass) of $Ln_3IrO_7$ will be anticipated.

Meanwhile, direct surface deprotonation of crucial intermediates on $IrO_x$ is rather difficult at high proton concentration in acidic medium. Introducing additional surface proton acceptors helps to optimize deprotonation pathway of OER intermediates and accelerate reaction kinetics, but probably at the risk of covering pristine active sites[7,9,17–21]. Bridging oxygen ($O_{bri}$) can help with stabilizing oxo intermediates by proton transfer, potentially acting as proton acceptors[22]. However, highly active and under-coordinated $O^{(II-\delta)-}$ species shows strong hydrogen adsorption, thus hindering its further deprotonation

and OER kinetics. Incorporating foreign atoms in $Ir-O^{(II-\delta)-}-Ir$ can effectively modulate hydrogen adsorption of $O_{bri}$, similar to the case in Brønsted-type solid-acid catalysts[23,24]. Considering the crystal structure of $Ln_3IrO_7$, the multi-valent nature of Mo, the modulation of Brønsted acidity by Mo and the acidic stability of molybdenum oxide[4,5,23], it is expected that substitution of Ir in $Ln_3IrO_7$ with Mo can realize controllable surface reconstruction towards desired $Ir-O^{(II-\delta)-}-Mo$ active species with reduced Ir consumption and optimized electronic configuration.

In this work, we use the high-valence Mo modulated $Pr_3Ir_{1-x}Mo_xO_7$ ($x$Mo-PIO) as model to trigger the directional surface reconstruction by activating lattice oxygen and cations, forming self-terminated $Ir-O_{bri}-Mo$ active species. Combining theoretical and experimental approaches, we reveal that Mo substitution contributes to accelerated surface reconstruction due to optimized Ir–O covalency and more prone dissolution of Pr. The improved durability results from the increase of Mo oxidation state for charge compensation to fierce Pr dissolution, which effectively avoids excessive loss of lattice oxygen. Particularly, foreign Mo survives in such directional surface reconstruction that highly active $Ir-O_{bri}-Mo$ species is generated. The remaining $MoO_x$ species in surface reconstruction layers promote the deprotonation process as proton acceptors, resulting in final enhancement in electrocatalytic activities. The optimal 0.2Mo-PIO only requires an overpotential ($\eta$) of 259 mV to reach 10 mA $cm_{geo}^{-2}$ and exhibits excellent stability for over 200 h. This work offers a facile strategy to regulate the favorable surface reconstruction toward both

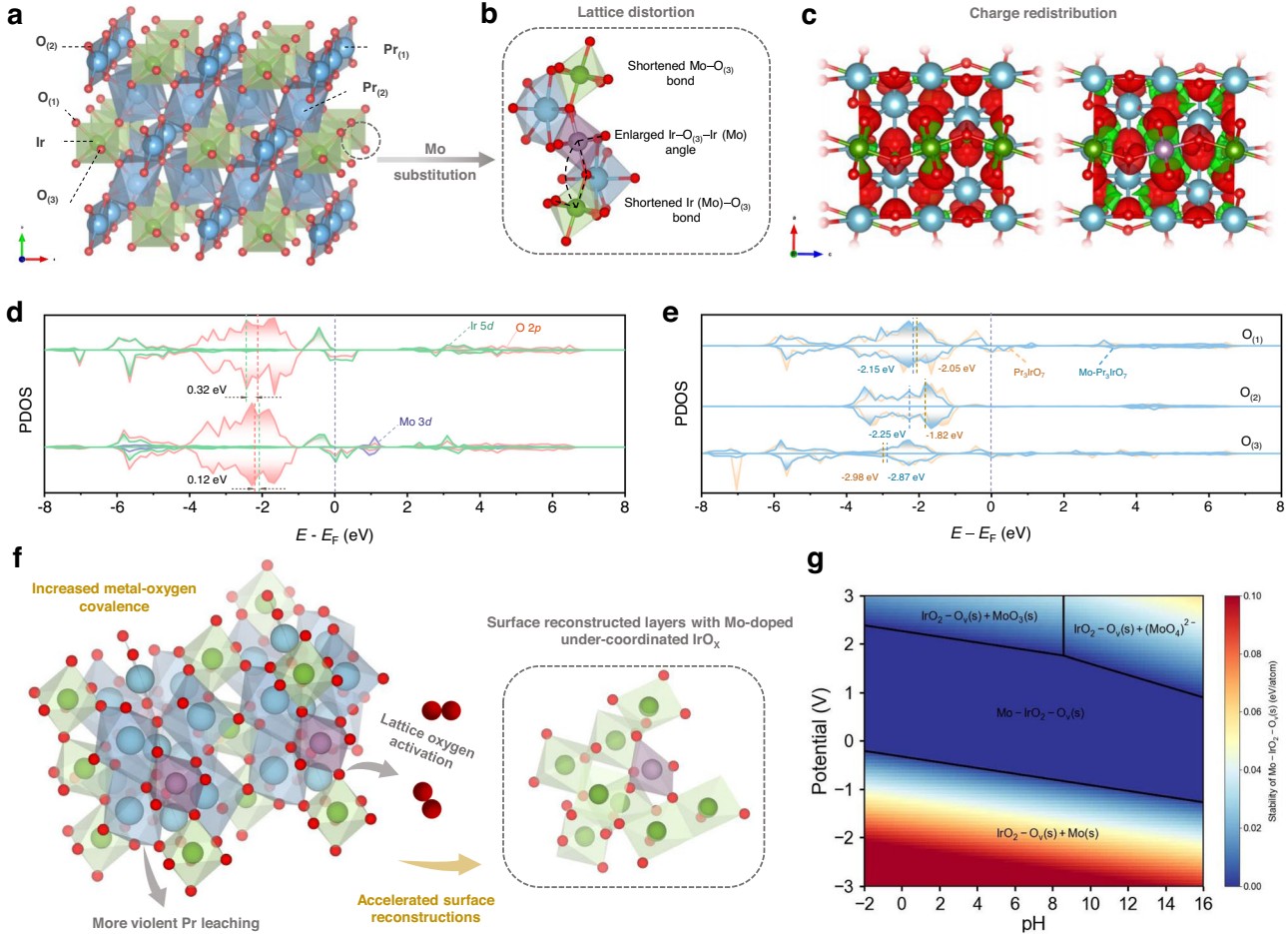

**Fig. 1 | DFT calculations on Mo-redirected surface reconstruction of $Pr_3IrO_7$.** **a** Crystal structures of Weberite type $Pr_3IrO_7$. **b** Lattice distortion induced by Mo substitution. **c** Charge density difference for $Pr_3IrO_7$ (left) and $Mo-Pr_3IrO_7$ (right). Color code: Pr (blue), Ir (green), Mo (purple), and O (red). Red and green shadows represent charge accumulation and depletion, respectively. **d** PDOS of Ir $5d$, Mo $3d$, and O $2p$ orbitals for $Pr_3IrO_7$ (top) and $Mo-Pr_3IrO_7$ (bottom). **e** PDOS of different kinds of lattice oxygen. **f** Schematic illustration for the accelerated surface reconstruction upon Mo substitution. **g** Pourbaix diagram of $Mo-IrO_2-O_v$.

highly active and stable species, inspiring the development of efficient OER electrocatalysts with reduced consumption of precious metals for PEM applications.

## Results

### DFT understanding Mo-redirected surface reconstruction of $Pr_3IrO_7$

The physical origin of surface reconstruction of metal oxide is resulted from high metal–oxygen covalency[25,26]. Based on the flexible crystal structure and characteristic band structure of $Pr_3IrO_7$, high-valent Mo is doped into $Pr_3IrO_7$ to accelerate surface reconstruction and obtain active reconstructed layers (Fig. 1). Using density functional theory (DFT) calculations, we first explored the probabilities of Mo substitution for different metal sites and the corresponding geometric and electronic structure of Mo-doped $Pr_3IrO_7$. As seen, the formation energy is more favorable for Ir-site substitution (0.25 eV) than that for Pr-site substitutions (7.94 eV for $Pr_{(1)}$, 9.57eV for $Pr_{(2)}$), demonstrating that Mo atoms are preferred to occupy six-coordinated Ir sites (Supplementary Fig. S1 and Supplementary Table S1). Besides, obvious lattice distortion is observed (Fig. 1b, detailed bond parameters are labeled in Supplementary Fig. S2). The enlarged Ir–$O_{(3)}$–Ir (Mo) angles along $c$-axis imply possibly increased Ir (Mo)–O covalency and the consequent activation of lattice oxygen atoms[16]. Charge density difference also verifies the charge redistribution upon Mo substitution (Fig. 1c). The difference was implemented by subtracting the partial charge density of the Ir, Mo, Pr, and O atoms from the whole charge density of the (Mo−) $Pr_3IrO_7$. An obvious charge accumulation (red shadows) is observed around Ir sites accompanied with charge depletion (green shadows) around Mo sites. The Bader charge analysis further quantifies that the Ir atoms of Mo−PIO show the lower (0.16 | e|) charge depletion than that of PIO, indicating the electrons are transferred from Mo to Ir (Supplementary Table S2).

The charge-transfer energy was further calculated to estimate metal–oxygen covalency[27]. Figure 1d shows the calculated projected density of states (PDOS) of bulk PIO and Mo−PIO. The O 2$p$-band center and Ir 5$d$-band center are displayed by integrating the PDOS and the difference between them denotes the charge-transfer energy. As depicted in Fig. 1d, the lower charge-transfer energy indicates an increased Ir–O covalency of Mo−PIO compared with PIO, manifesting increased activity of lattice oxygen (Supplementary Table S3). PDOS of different kinds of O atoms ($O_{(1)}$, $O_{(2)}$, $O_{(3)}$) in PIO and Mo−PIO were conducted to give more accurate determination (Fig. 1e). After Mo substitution, the O 2$p$-band center of $O_{(1)}$, $O_{(2)}$ decreases relative to Fermi level, while that of $O_{(3)}$ increases. It can be inferred that the decrease of O 2$p$-band center is mainly caused by $O_{(1)}$ and $O_{(2)}$, and the enlarged Ir–O covalency, which facilitates the reactivity of lattice oxygen, mainly origins from the increase of Ir 5$d$-band center (Fig. 1d and Supplementary Table S3). Consequently, the more violent evolution of surface structures would occur during electrochemical process.

Vacancy formation energies were further calculated to demonstrate the possible reconstruction behavior of cations and lattice oxygen. $Pr_{(1)}$ and $Pr_{(2)}$ vacancy formation energies have decreased significantly, indicating the easier escape of Pr from bulk under OER conditions (Supplementary Table S4). As shown in Supplementary Table S5, $O_{(1)}$ vacancies are inclined to form in PIO, while after Mo substitution, vacancy formation energies have decreased for $O_{(3)}$ (from 2.83 to 2.66 eV) and $O_{(1)}$ in Mo−$O_{(1)}$ bonds (from 2.73 to 2.64 eV). In addition, leaching of Pr is further accelerated upon loss of lattice oxygen. As seen, Pr vacancy formation energies further decrease when oxygen atoms are removed from the lattice (9.58 eV for pristine samples compared with 8.50 eV and 9.37 eV after the formation of $O_{(1)}$ and $O_{(3)}$ vacancies in Mo-doped $Pr_3IrO_7$), indicating probable surface reconstruction process mediated by activation of lattice oxygen coupled metal dissolution (Supplementary Table S6). These results suggest that moderate substitution of Ir by Mo in $Pr_3IrO_7$ results in lattice distortion and charge redistribution, where the activated Pr–O bond and increase in Ir–O covalency potentially contribute to faster and favorable surface reconstruction during OER (Fig. 1f).

Previous studies show that doping elements in doped iridates generally suffer drastic dissolution during surface reconstruction, thus failing to participate in OER within reconstructed $IrO_x(OH)_y$ layers[12,28], which exhibit unfavorable deprotonation because of strong hydrogen adsorption of $O^{(II-\delta)-}$. Differently, formation of Ir–$O_{bri}$–Mo active species is anticipated here due to acid stability of molybdenum oxide, which is further confirmed by the Pourbaix diagram (Fig. 1g). When the applied potential lies within OER potential window, a wide domain of thermodynamic stability is observed for Mo−$IrO_2$−$O_v$ in acid region. Electrochemical stability of the Mo−$IrO_2$−$O_v$ indicates that Mo is expected to remain in reconstructed layers for regulating OER process, as evidenced by the following experimental results.

### Synthesis and structural characteristics of Mo-doped $Pr_3IrO_7$

In light of the above results, $x$Mo-PIO ($x$ = 0, 0.1, 0.2, 0.4) were synthesized by a solid-state method (see "Methods" for details)[11]. Powder X-ray diffraction (XRD) patterns confirm an orthorhombic phase of as-synthesized $x$Mo-PIO with space group $Cmcm$, well matched with $Pr_3IrO_7$ (PDF # 97-008-6473) (Fig. 2a). A clear shift of (202) diffraction peak toward lower angles is observed with increased Mo substitution (Fig. 2a, right panel), indicating a crystal lattice expansion owing to larger ion radius of $Mo^{VI/V}$ than that of $Ir^V$ (Supplementary Table S7). Meanwhile, the intensity of $Pr_3IrO_7$ (112) diffraction peak gradually weakens until a new peak of a relatively small impurity appears nearby which can be indexed to (222) peak of $Pr_2O_{3.33}$ (PDF # 97-064-7294), as shown in 0.4Mo-PIO. A small amount of high-valence $Mo^{VI/V}$ substitution results in a decrease in the average oxidation state of Ir to realize charge compensation. When doped with excess $Mo^{VI/V}$, however, Pr segregates to the surface forming praseodymium oxides for further charge balancing (Supplementary Fig. S3). Scanning electron microscopy (SEM) images illustrate that the $x$Mo-PIO crystallized into irregular and large polyhedral particles about several micrometers because of the high calcination temperature (Supplementary Fig. S4). Less Mo substitution ($x$ = 0.1 and 0.2) is easier to eliminate residual strain induced by lattice distortion to avoid agglomerate. This is further supported by the larger Brunauer–Emmett–Teller (BET) surface areas based on $N_2$ adsorption/desorption measurements (Supplementary Fig. S5 and Supplementary Table S8). Indeed, electrocatalysts with larger surface areas and smaller particle size are beneficial for exposing more active site to catalyze surface reactions.

Figure 2b, c present typical high-resolution transmission electron microscopy (HRTEM) images of PIO and 0.2Mo-PIO, respectively. The interplanar spacing of about 0.308 nm and 0.549 nm with a dihedral angle of 55.89° coincide well with (220) and (200) facets of orthorhombic phase $Pr_3IrO_7$. In addition, the selected-area electron diffraction (SAED) pattern along [001] zone axis (insets) of PIO shows a discrete reciprocal lattice, confirming a highly ordered single-crystalline nature. And 0.2Mo-PIO exhibits similar crystallization characteristics with homogeneous dispersion of Mo in the lattice as revealed by energy dispersive X-ray spectroscopy (EDS) elemental mapping (Supplementary Fig. S6). All these results demonstrate that moderate amount Mo (x ≤ 0.2) atoms are successfully doped into $Pr_3IrO_7$ lattice and the weberite characteristics of Mo-$Pr_3IrO_7$ are well maintained.

X-ray photoelectron spectroscopy (XPS) measurements were performed to study surface electronic properties of $x$Mo-PIO. Ir 4$f$ core-level spectra exhibit two broad signals of 4$f_{5/2}$ and 4$f_{7/2}$ with evident variation in peak shapes and full width at half maximum (FWHM), manifesting changes in Ir chemical environment by Mo substitution (Fig. 2d). Two doublets centered at 63.7 eV and 66.7 eV, and 62.5 eV and 65.5 eV can be ascribed to $Ir^V$ and $Ir^{<V}$ species, respectively[29–31]. The gradually increased content of $Ir^{<V}$ with increasing Mo substitution

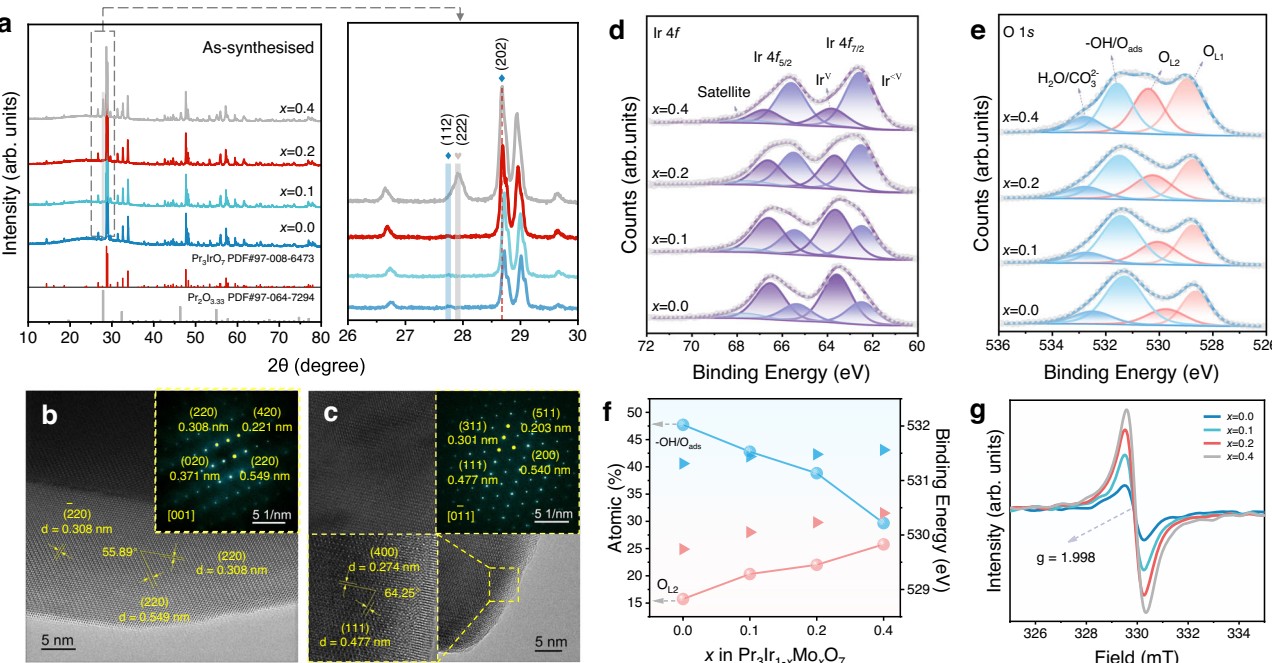

**Fig. 2 | Structural characterizations of *x*Mo-PIO. a** Powder XRD patterns. **b**, **c** HRTEM images of PIO (**b**) and 0.2Mo-PIO (**c**). Insets: corresponding SAED patterns. The relevant interplanar spacing and dihedral angles are labeled. **d**, **e** XPS spectra of Ir 4*f* (**d**) and O 1*s* (**e**) core-level with peak fitting results. **f** Atomic ratio comparison of fitting results and the binding energy changes for −OH/O$_{ads}$ (blue ball and blue triangle) and O$_{L2}$ (red ball and red triangle). **g** EPR results.

indicates a lower average Ir oxidation state owing to difference in electronegativity ($\chi$) of Ir ($\chi = 2.2$) and Mo ($\chi = 2.16$) for charge redistribution along Ir−O$_{(3)}$−Mo chains, consistent with Bader charge analysis (Supplementary Table S9). The less positive-charged Ir$^{<V}$ species conduce to increased metal−oxygen covalency and activation of lattice oxygen, as predicted by theoretical calculations[8,32–34]. O 1*s* spectra were fitted into four components (Fig. 2e and Supplementary Note 2)[29,35]. The binding energies of O$_{L2}$ (highly oxidized surface lattice oxygen species, O$_2^{2-}$/O$^-$) and −OH/O$_{ads}$ species were analyzed in detail as well as their content variations (Fig. 2f and Supplementary Table S10). Mo doping renders slightly positive-shifted binding energies for both of O$_{L2}$ and −OH/O$_{ads}$ accompanying higher contribution from O$_{L2}$ but lower from ·OH/O$_{ads}$, indicating highly activated surface with a large amount of electrophilic oxygen species[36–38]. Electrophilic O$_{L2}$ species was further evidenced by electron paramagnetic resonance (EPR) due to its sensitivity to unpaired electrons trapped by lattice defect. The stronger signal intensities at $g = 1.998$ for Mo-doped samples than that of PIO agree well with the results from XPS analysis (Fig. 2g)[39,40].

## Electrochemical performance evolution upon surface reconstruction

OER performance was assessed using a typical rotating disk electrode (RDE) technique based on a conventional three-electrode configuration in 0.1 M HClO$_4$ electrolyte under ambient conditions. The as-synthesized *x*Mo-PIO experienced gradually increased OER activity over continuous electrochemical cycling (Supplementary Fig. S7), which demonstrate a dynamic evolution of surface structure and the generation of more active species under electrochemical conditions. The potential changes to reach a current density of 10 mA cm$_{geo}^{-2}$ over the first 20 cycles clearly shows the superior OER activity of Mo-doped samples compared with pure Pr$_3$IrO$_7$ (Fig. 3a, samples delivering a steady polarization curve are denoted as *x*Mo-PIO-post hereafter). Specifically, 0.2Mo-PIO shows higher initial OER activity because of enlarged Ir–O covalency. Besides, a faster decrease in potential for Mo-doped samples compared with PIO can be demonstrated, which indicates a more drastic and favorable surface reconstruction due to easier

Pr dissolution and lattice oxygen activation. The optimal 0.2Mo-PIO-post only requires an overpotential ($\eta$) of 259 mV to reach 10 mA cm$_{geo}^{-2}$, much lower than that of PIO-post (310 mV). As shown in Supplementary Fig. S8 0.2Mo-PIO-post gives a mass activity of 415 A·g$_{Ir}^{-1}$ at 1.52 V, about 4.3 and 88.3 times higher than that of PIO-post (97 A·g$_{Ir}^{-1}$) and commercial IrO$_2$ (4.7 A·g$_{Ir}^{-1}$). The iridium mass activity is also compared with reported Ir-based catalysts in acidic media (Supplementary Table S11), demonstrating the high mass activity of 0.2Mo-PIO-post among reported Ir-based catalysts. And OER performance tests conducted in 0.5 M H$_2$SO$_4$ further reveal that 0.2Mo-PIO-post is among the most active electrocatalysts for OER in acid (Supplementary Fig. S9 and Supplementary Table S12).

The different surface redox properties induced by Mo substitution during electrochemical reconstruction was further traced from cyclic voltammetry (CV) measurements (Fig. 3b). The redox peak at around 1.38 V related with lattice oxygen oxidation process is observed for both samples at the first cycle due to their high lattice oxygen reactivity as demonstrated from band structures. Two successive oxidation peaks at about 0.9 V and 1.2 V in PIO-post correspond to Ir$^{III/IV}$ and Ir$^{IV/V}$ species, suggesting the formation of Ir-enriched surfaces with highly active species[12,41,42]. As for 0.2Mo-PIO-post, however, profile of CV curves is broader and the oxidation peak at 0.98 V, generally considered as deprotonation of two-coordinated bridge oxygen, is much more dominant, indicating a deprotonation process of active oxygen intermediates regulated by Mo[22,43,44]. Meanwhile, the higher current density of 0.2Mo-PIO-post indicates the larger electrochemically active surface area (ECSA) and the increased degree of surface reconstruction. Kinetic currents normalized to ECSA provide a comparison of intrinsic activity; the higher current density of 0.2Mo-PIO-post explicitly shows the superior intrinsic activity modified by Mo substitution (Fig. 3c, marked as squares; Supplementary Figs. S10 and S11a, b; the corresponding rough factors are listed in Supplementary Table S13), same as the trend obtained from geometric area-normalized current densities (Fig. 3c, marked as columns and Supplementary Fig. S11c). Tafel slope derived from the geometric area-normalized activity of 0.2Mo-PIO-post (50.52 mV dec$^{-1}$) is noticeably smaller than that of

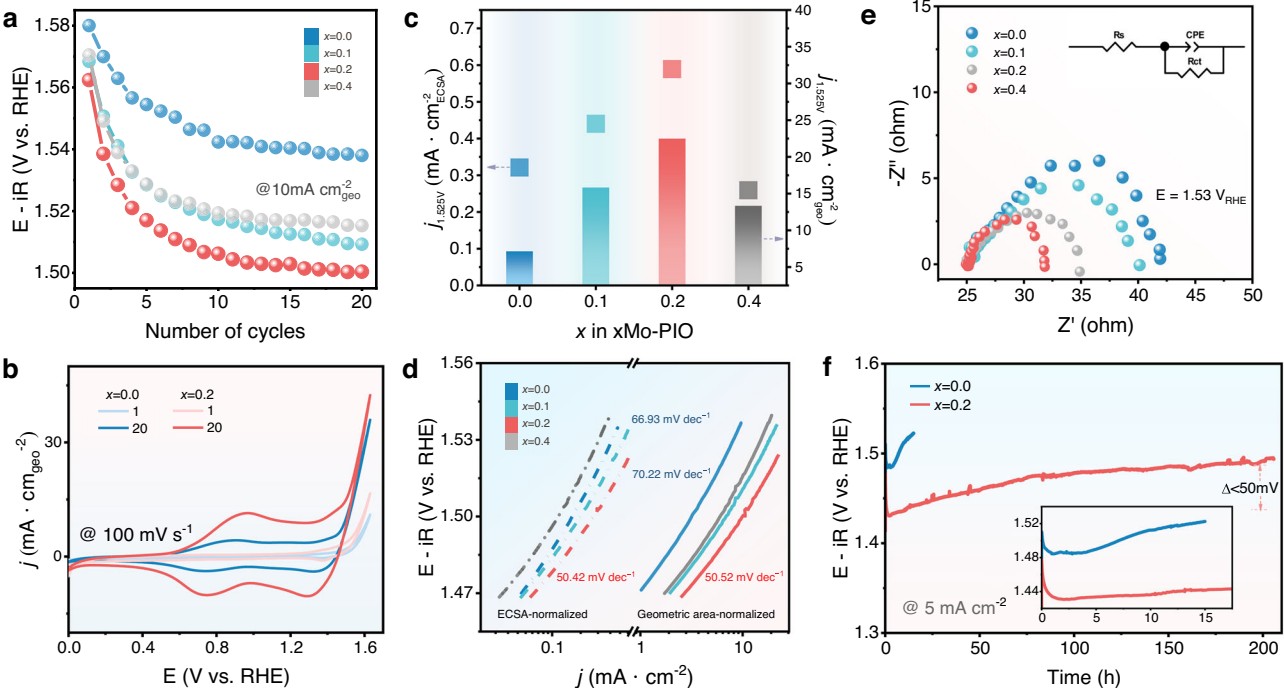

**Fig. 3 | Electrochemical performance of *x*Mo-PIO during surface reconstruction. a** Evolution of overpotential to reach 10 mA cm$_{geo}^{-2}$ from LSV curves over the first 20 cycles at 5 mV s$^{-1}$. **b** CV curves recorded at the 1st and 20th cycles in 0.1 M HClO$_4$ electrolyte at a scan rate of 100 mV s$^{-1}$. **c** Comparisons of ECSA- and geometric area-normalized current densities at 1.525V$_{RHE}$ post 20 LSV scans. **d** Tafel plots based on ECSA- and geometric area-normalized LSV curves post 20 LSV scans. **e** Nyquist plots at 1.53 V$_{RHE}$, and the inset shows the electrical equivalent circuit. The correspond R$_{ct}$ is 22.67 Ω, 15.95 Ω, 6.709 Ω, and 10.23 Ω for *x* = 0.0, 0.1, 0.2, and 0.4, respectively. **f** Chronopotentiometry curves at 5 mA cm$^{-2}$ in 0.1 M HClO$_4$ electrolyte. Inset shows the potential changes over the first 20 h.

PIO-post (70.22 mV dec$^{-1}$), consistent with results obtained from ECSA-normalized plots (Fig. 3d and Supplementary Table S14), further confirming the optimized kinetics of reconstructed species with Mo substitution. The distinction in Tafel slopes implies probably modified OER pathway, as discussed in the following parts[45–48]. Interfacial charge transfer was assessed via electrochemical impedance spectroscopy (EIS) measurements. The decreased semicircles in Nyquist plots clearly demonstrate a facilitated charge-transfer process of 0.2Mo-PIO-post with smaller charge-transfer resistance (R$_{ct}$) (6.71 Ω) compared with PIO-post (22.67 Ω) (Fig. 3e and Supplementary Table S15). Besides, 0.2Mo-PIO exhibits excellent stability with an overpotential increment less than 50 mV at 5 mA cm$^{-2}$ after 200 h, while PIO deteriorates quickly within 20 h (Fig. 3f). Chronopotentiometry curves at 20 mA cm$^{-2}$ manifests the better stability of 0.2Mo-PIO (overpotential increment less than 30 mV after 140 h) than commercial IrO$_2$ (Supplementary Fig. S12), as further verified by the higher stability number (S-number) of $2.1 \times 10^8$ (see "Methods" for details, comparisons of stability number with other Ir-based electrocatalysts are listed in Supplementary Table S16). These results validate that an appropriate concentration of Mo doping can accelerate surface reconstruction process and modify the final active species to realize significantly improved OER performance.

## Mo-buffered charge compensation during reconstruction

To better understand the surface evolution and the role of Mo during reconstruction, HRTEM and XPS for PIO and 0.2Mo-PIO were performed after a certain number of electrochemical cycles. The [00$\bar{1}$]-oriented HRTEM image of PIO and [11$\bar{2}$]-oriented of 0.2Mo-PIO after two cycles show well-ordered crystalline, proved by the clear SAED diffraction patterns and lattice fringes (Fig. 4a, b, left panels). Sparse and uniformly dispersed particles appear on the surface without discernible depth but denser for 0.2Mo-PIO, manifesting quicker surface evolution because of increased Ir–O covalency and the consequent

lattice oxygen activation[9,13,49]. As seen, O$_{L1}$ and O$_{L2}$ in O 1s XPS spectra almost disappear, especially for 0.2Mo-PIO with greatly increased broad peak at around 532.4 eV (Fig. 5a). The steeper Pr/(Ir + Mo) fall for 0.2Mo-PIO (from 3.3 to 1.86) compared to PIO (from 2.9 to 1.94) indicates more drastic Pr leaching, corresponding to significantly decreased Pr vacancy formation energies upon Mo substitution and thermodynamic instability. (Supplementary Fig. S13 and Supplementary Tables S17 and S18). Besides, Ir 4f XPS spectra of two samples show noticeable differences. There is still Ir$^V$ species remaining in PIO, which completely disappear but accompanying formation of abundant Ir$^{III}$ species in 0.2Mo-PIO (Fig. 5b, c and Supplementary Note 3)[30,31,50]. The oxidation state decrease of Ir in PIO is due to the more kinetically sluggish regeneration of surface lattice oxygen compared with its activation[9,51]. However, the obvious positive shift of Mo 3d peaks manifest that the decrease in Ir oxidation state after two cycles is associated not only with lattice oxygen activation but with charge redistribution induced by Mo (Fig. 5d)[42]. Obviously, Mo substitution facilitates Pr dissolution and lattice oxygen activation, contributing to accelerated surface reconstruction and a faster improvement in OER activity, consistent with calculation results.

On the other hand, the consequent under-coordinated cation sites adjacent to oxygen vacancies on such metastable surface are more likely to dissolve under oxidation potentials[8,9,52]. After ten cycles, bright and discrete diffraction patterns along [01$\bar{1}$] zone axis for PIO and [$\bar{1}$11] zone axis for 0.2Mo-PIO demonstrate the well-preserved bulk crystalline (Fig. 4a, b, middle panel). Closer particle distribution is observed in PIO with a diameter of roughly 3.5 nm (Fig. 4a, middle panel), which can be identified as IrO$_2$ nanoparticles (marked with orange in the insets). Distinctively, surface structure evolution of 0.2Mo-PIO is far more apparent with a depth of approximately 6.5 nm (Fig. 4b, middle panel). The deeper surface reconstruction may be attributed to more severe Pr leaching as verified from the lower Pr/(Ir + Mo) value (Supplementary Tables S17 and S18), which is also

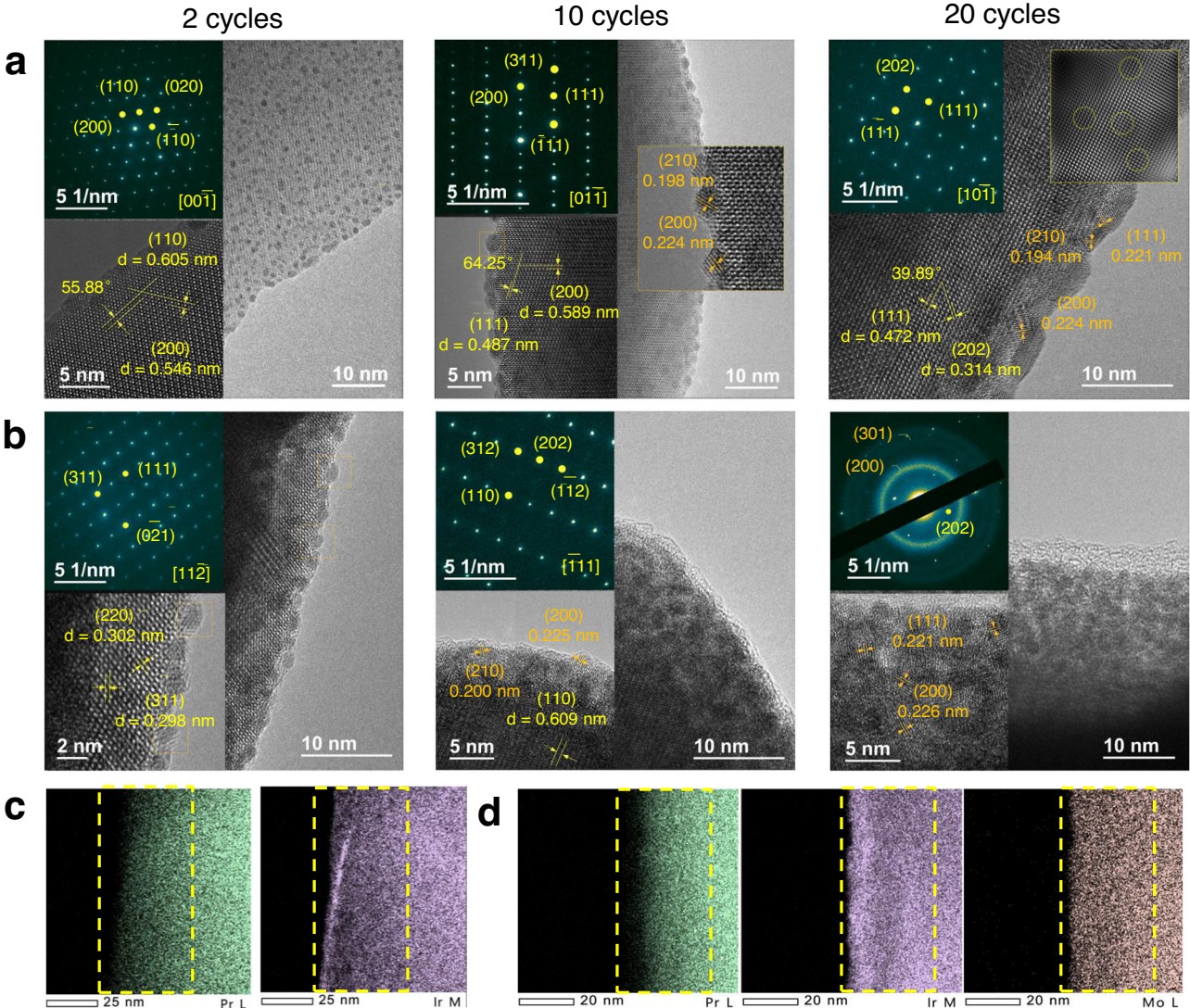

**Fig. 4 | Dynamic surface evolution and deformation of bulk phase in the electrochemical process. a, b** TEM images of PIO (**a**) and 0.2Mo-PIO (**b**) at varied electrochemical cycles. Left, middle, and right panels correspond to 2, 10, and 20 cycles, respectively. The insets at the top left in (**a, b**) show the SAED patterns for each corresponding HRTEM images. The relevant interplanar spacing and dihedral angles are labeled. **c, d** EDS elemental mappings of Pr, Ir, and Mo elements for PIO (**c**) and 0.2Mo-PIO (**d**) post 20 LSV cycles. The yellow dashed lines indicate the metal enrichment and consumption at near-surface regions.

corroborated by the DFT calculations where lattice oxygen evolution further facilitates leaching of Pr. The positive-shifted O 1s peak with a higher content of -OH/$O_{ads}$ species in PIO demonstrates charge compensation from further lattice oxygen loss when suffering from violent cation dissolution, consistent with the emergence of $Ir^{III}$ species (Fig. 5a, b)[9,36]. However, inspection of Mo 3d XPS spectra of 0.2Mo-PIO exhibit a continuously increased oxygen state, so as to compensate charge imbalance induced by fierce Pr leaching, which prevents excessive loss of active oxygen species (Fig. 5d). The buffering of Mo explains the more content of $Ir^{III}$ species but negative-shifted O 1s peaks of 0.2Mo-PIO (Fig. 5a, c).

Despite the crystalline state of bulk structure confirmed by clear diffraction patterns along [10$\bar{1}$] zone axis in PIO after 20 cycles, the precipitated $IrO_2$ particles on the surface have grown to more compact and thicker bricks with a depth around 4 nm. The disorderly zone distribution with different luminance in filtered HRTEM image reveals bulk defects arising from the migration of Ir from bulk to surface (Fig. 4a, right panel)[8]. The continuously decreased Ir oxidation state and the simultaneously positive-shifted O 1s peaks demonstrate further variation of surface oxygen species, implying an unstable kinetics

of surface oxygen recycles (Fig. 5a, b)[53,54]. Differently, the O 1s and Ir 4f XPS spectra after 20 cycles exhibit negligible variation compared with those after 10 cycles of 0.2Mo-PIO (Fig. 5a, c). Mo 3d XPS spectra shows that Mo valence increases for opportune charge compensation through surface Ir–$O_{bri}$–Mo (Fig. 5d), thereby effectively impeding structure collapse and loss of active species. The obviously attenuated patterns of bulk diffraction and the emergence of extra diffraction rings suggest an ongoing surface rearrangement towards mixed phases of crystalline and amorphous (Fig. 4b, right panel), as indicated by the continually decreasing Pr/(Ir + Mo) values (Supplementary Tables S17 and S18). Obviously, more fierce Pr depletion and Ir enrichment layers at edge is found in 0.2Mo-PIO, indicating a deeper reconstruction (Fig. 4c, d). And notably, compared with Pr, Mo in near-surface regions shows relatively even distribution without obvious depletion, indicating that Mo is remaining in surface reconstruction layers, which is responsible for the better electrocatalytic performance.

We confirmed that the electrochemical activation of outmost surface lattice oxygen and the initial dissolution of Pr are responsible for surface reconstruction. More fierce leaching of Pr and promoted

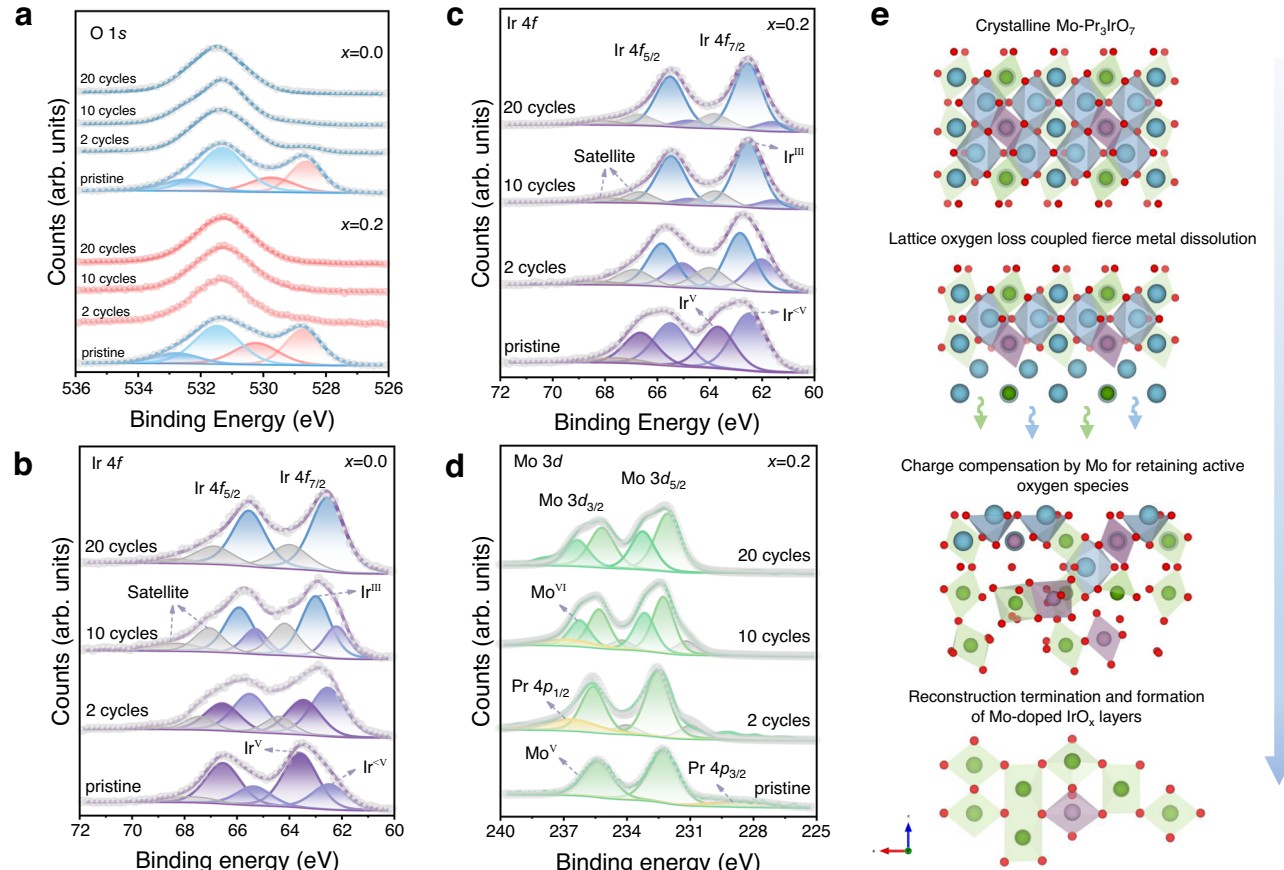

**Fig. 5 | Surface chemical states evolution in reconstruction. a** XPS spectra of O 1*s* core-level with peak fitting results for PIO and 0.2Mo-PIO at varied electrochemical cycles. **b**, **c** XPS spectra of Ir 4*f* core-level with peak fitting results for PIO (**b**) and 0.2Mo-PIO (**c**) at varied electrochemical cycles. **d** XPS spectra of Mo 3*d* core-level with peak fitting results for 0.2Mo-PIO at varied electrochemical cycles. **e** Schematic illustration of electrochemical surface reconstruction over Mo-Pr₃IrO₇.

lattice oxygen reactivity induced by Mo substitution contribute to deeper reconstruction with more exposure of catalytically active sites. Besides, charge compensation to Pr dissolution via increase of Mo oxidation state during reconstruction effectively avoids excessive loss of lattice oxygen and enables opportune reconstruction into self-terminated Ir–O$_{bri}$–Mo species. (Fig. 5e).

**Mo-induced strong Brønsted acidity of O$_{bri}$ in BOAD mechanism**
Despite the similar oxidation state of Ir species for PIO-post and 0.2Mo-PIO-post, the significantly promoted OER performance suggests crucial role of Mo in regulating the electronic structure in reconstruction layers. Generally, if the potential determining step (PDS) involves decoupled proton-electron transfer process, OER activities will be very sensitive to proton concentration in the electrolyte[55,56]. pH dependence measurements and deuterium isotopic labeling experiments were carried out to investigate the dependence of OER kinetics on proton activity[57,58]. Results of LSV measurements at different pH (pH = 0.79, 1.01, 1.17, 1.41, and 1.61) for PIO-post and 0.2Mo-PIO-post are shown in Fig. 6a. And the proton reaction order $\rho^{RHE}$ derived from the partial derivation of current density (in log scale) at specific voltage (1.53 V vs. RHE here) with respect to pH values are given in Fig. 6b. Obviously, PIO-post is more sensitive to proton concentration with a larger $\rho^{RHE}$ of 0.41, indicating that deprotonation process is involved in PDS with decoupled electron transfer process. However, $\rho^{RHE}$ of 0.2Mo-PIO-post decreases to 0.19, suggesting that remaining Mo in surface reconstruction layers has facilitated the proton transfer to more favorably concerted proton coupled electron transfer process[56]. OER activities were also tested in proton (0.1 M HClO₄ in H₂O) and deuterium (0.1 M HClO₄ in D₂O) electrolytes to

investigate the kinetic isotope effect (KIE) (Fig. 6c and Supplementary Note 4)[59–61]. KIE values were obtained based on the ratio of OER current density for hydrogen to that for deuterium at the same overpotential (Fig. 6d). PIO-post exhibits an obviously larger KIE value compared with 0.2Mo-PIO-post, demonstrating a larger dependence of OER process on proton activity. Consequently, Mo in reconstruction layers contributes to the promoted deprotonation process, which is responsible for the faster OER kinetics than undoped. The deuterium isotopic experiments conducted in 0.5 M H₂SO₄ (in H₂O) and 0.5 M D₂SO₄ (in D₂O) also provided consistent results (Supplementary Fig. S14).

In situ electrochemical Raman spectroscopy was further employed to better decipher the critical role of Mo in reconstruction layers toward OER. As seen, characteristic peaks at ~134 cm⁻¹, ~259 cm⁻¹, ~313 cm⁻¹, ~342 cm⁻¹, ~586 cm⁻¹ for pristine samples have vanished with only one peak left at ~653 cm⁻¹ and a new peak at ~460 cm⁻¹ (Ir−O stretch vibrations) (Fig. 6e, OCP; Supplementary Fig. S15, Supplementary Note 5)[62–64]. Particularly, Mo−O characteristic vibrations at 770 cm⁻¹ (vibration mode) and ~850 cm⁻¹ (stretching mode) in 0.2Mo-PIO-post demonstrate that Mo is survived from the drastic surface reconstruction. When the applied potential increases from 0.7 to 0.9 V, the peak intensity at ~520 cm⁻¹ (Ir−O stretching vibrations) associated with Ir$^{III}$ decreases accompanying increased peak intensity of at ~460 cm⁻¹ (Ir−O stretching vibrations) related to Ir$^{IV}$ species in 0.2Mo-PIO-post. This transition demonstrates oxidation of Ir$^{III/IV}$, consistent with the redox peaks in CV plots (Fig. 3b). Besides, Mo−O vibration mode at around 770 cm⁻¹ gradually shifts to lower frequencies, implying a probable deprotonation process assisted by Mo, which explains prominent peak at ~0.98 V in CV cycles. In contrast, peak

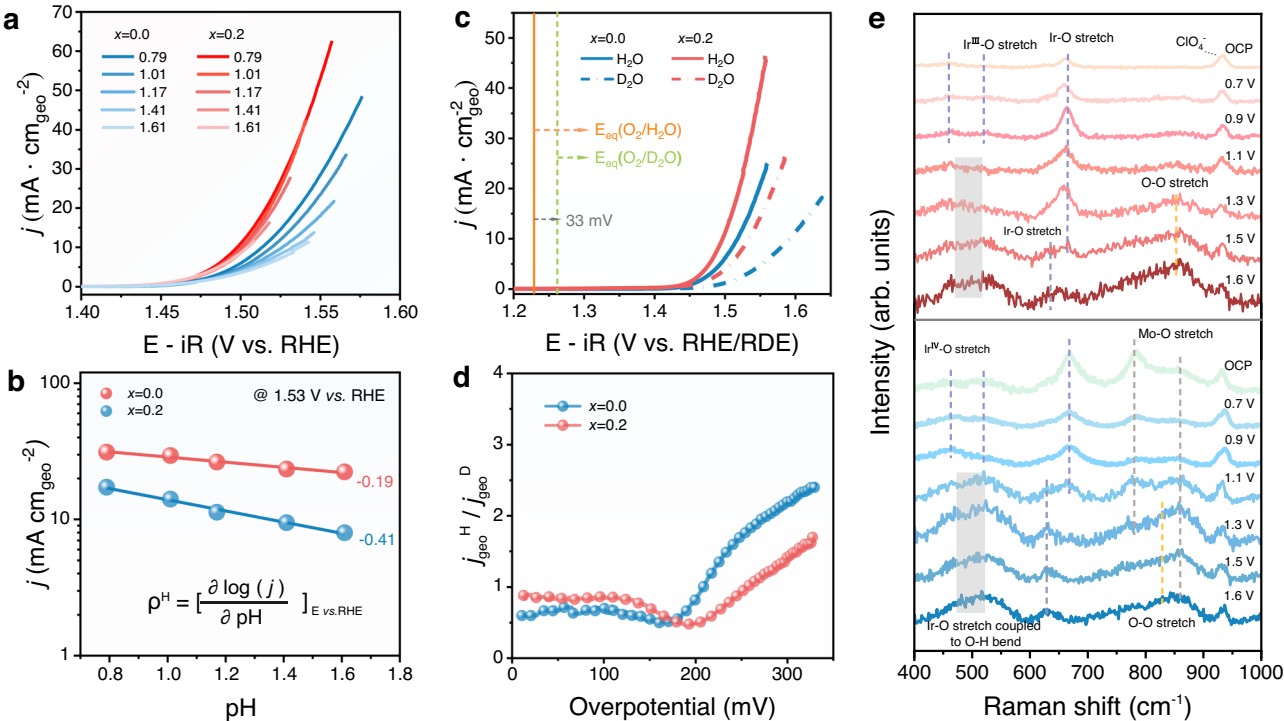

**Fig. 6 | Role of reconstructed Ir–$O_{bri}$–Mo species in promoting deprotonation.** **a** LSV curves for PIO-post and 0.2Mo-PIO-post in $HClO_4$ with varied pH. **b** OER current density at 1.53 $V_{RHE}$ plotted in log scale as a function of pH, from which the proton reaction orders ($\rho^H = \partial \log j/\partial pH$) were calculated. **c** LSV curves for PIO-post and 0.2Mo-PIO-post measured in 0.1 M $HClO_4$ prepared in $H_2O$ and $D_2O$. **d** KIE of

PIO-post and 0.2Mo-PIO-post. $j^H$ and $j^D$ are referred to the current density measured in 0.1 M $HClO_4$ prepared in $H_2O$ and $D_2O$ at the same overpotential, respectively. **e** In situ Raman spectra of PIO-post (red) and 0.2Mo-PIO-post (blue) in 0.1 M $HClO_4$ at varied potentials.

intensity at ~520 $cm^{-1}$ for PIO-post undergoes gradual increase and subsequent decrease in the same potential range. The emergence of such peak may arise from loss of lattice oxygen under oxidizing potentials, as corroborated by the negative shift of Ir–O vibrations at ~653 $cm^{-1}$. The following replenishing with hydroxyl and subsequent oxidizing leads to the positive shift of this peak. As potential increases from 0.9 to 1.6 V, Ir–O vibrations at ~660 $cm^{-1}$ has significantly weakened with the appearance of a new band at ~620 $cm^{-1}$ with negative shift because of weakened Ir–O bonds. Besides, a broadband associated with various Ir–O vibrations coupled to O – H vibrations appears at ~500 $cm^{-1}$(see ref. 62). However, there are noticeable difference in Raman peaks between two samples at same applied potential. Peak at ~620 $cm^{-1}$ for PIO-post is not recognized until the potential increases to 1.5 V. The asymmetry and broad peaks at 700–850 $cm^{-1}$ can be explained by O–O vibration at ~830 $cm^{-1}$ of Ir–OOH and possible Ir–O stretch vibration (out-of-plane), which is more prominent for 0.2Mo-PIO-post[42,62,63]. The significant distinction and unambiguous existence of Mo–O vibration modes confirm the influence of Mo on electronic structures in surface reconstruction layers, which accounts for the improved OER performance.

Above electrochemical and spectroscopic analysis demonstrate a deprotonation process promoted by Mo. DFT calculations were conducted based on the most thermodynamically stable $IrO_2$ (110) surface for further elucidating the role of Mo in reconstructed layers (Fig. 7a). There are three types of exposed sites on the surface: the coordinatively unsaturated Ir ($Ir_{cus}$), which is taken as active sites of adsorbed on-top oxygen ($O_{top}$) intermediates; the bridging oxygen ($O_{bri}$), coordinated with two fully coordinated bridge Ir ($Ir_{bri}$); the three-fold coordinated oxygen ($O_{3f}$), bonded with two $Ir_{cus}$ and one $Ir_{bri}$. The Brønsted acidity of $O_{bri}$ is a determined parameter for evaluating the deprotonation process. Substitution parent metal with of lower electronegativity elements would lead to shift of the electron density from

ligand to parent metal and further favorable hydrogen desorption on ligand[65,66]. Considering the modulation of the Brønsted acidity by Mo[23], [$MoO_6$] octahedrons were inserted to form a Mo–$O_{bri}$–Ir site (Supplementary Fig. S16). Besides, bridging-oxygen vacancies were introduced to coincide with XPS results aforementioned. The adsorption energy ($E_{ads}$) of hydrogen atoms on two types of $O_{bri}$ sites, Ir–$O_{bri}$–Ir in $IrO_2$-$O_v$ and Mo–$O_{bri}$–Ir in Mo-doped $IrO_2$-$O_v$ were firstly examined. Ir–$O_{bri}$–Ir shows fairly strong H adsorption with an $E_{ads}$ of −1.24 eV, whereas the adsorption strength is significantly weakened on Mo–$O_{bri}$–Ir with an $E_{ads}$ of −0.78 eV (Fig. 7b). This suggests more favorable deprotonation process due to the stronger Brønsted acidity of $O_{bri}$ upon Mo substitution, probably responsible for the less dependent of OER kinetics on proton activity for 0.2Mo-PIO-post as discussed above (Fig. 7c).

Given the critical role of $O_{bri}$ in assisting in water dissociation and oxo intermediates deprotonation, both conventional adsorbate evolution mechanism (AEM) pathway and bridging-oxygen-assisted deprotonation (BOAD) pathway are considered for $IrO_2$-$O_v$ and Mo-doped $IrO_2$-$O_v$. In AEM pathway, the electrochemical deprotonation of the oxo intermediates processes on $O_{top}$ site directly. While in BOAD pathway, a proton of the adsorbed intermediates is firstly chemically transferred onto a neighboring $O_{bri}$ site, then a coupled deprotonation accompanying electron transfer step takes place. Therefore, the elementary reaction step in AEM pathway is further disassembled in BOAD, and their corresponding free-energy differences ($\Delta G$) will be shared by two less values when the H adsorption strength of $O_{bri}$ is modest. Whereas in the case of $IrO_2$-$O_v$, proton transfer to $O_{bri}$ is preferred during deprotonation of *OOH intermediate because of the fairly strong proton adsorption capacity of Ir–$O_{bri}$–Ir site. Furthermore, PDS locates at deprotonation of OO$_{top}$*–OH$_{bri}$* with a large overpotential of 0.94 V (Supplementary Fig. S17, PDS is labeled by green star). As for Mo-doped $IrO_2$-$O_v$, with the introduction of BOAD

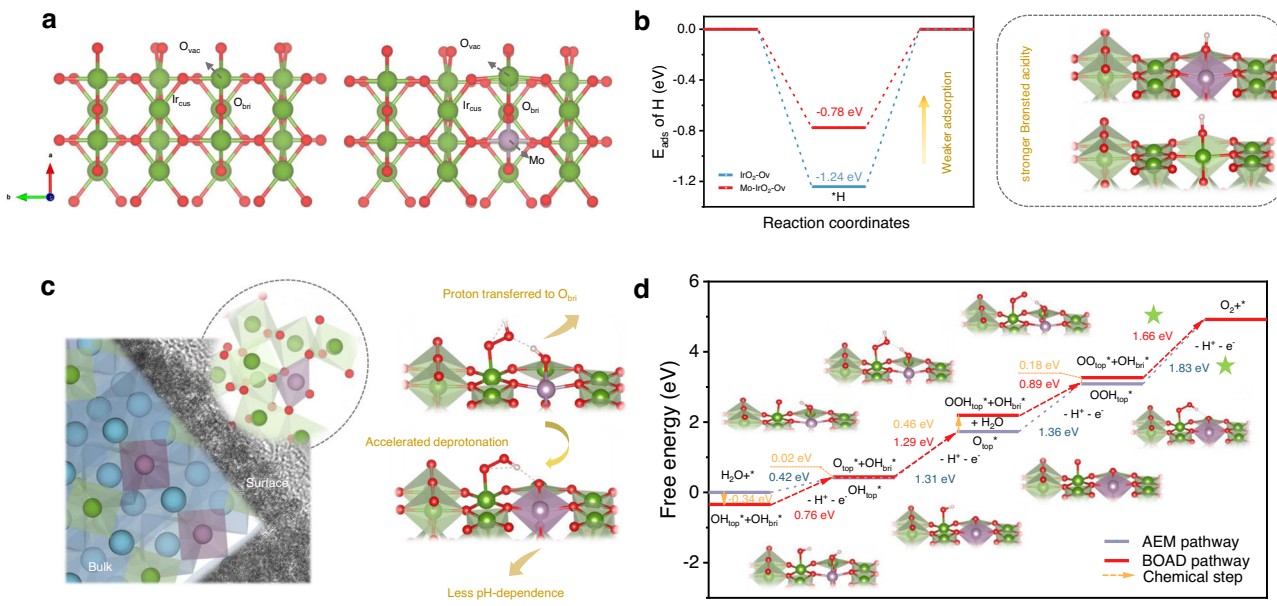

**Fig. 7 | Bridging-oxygen-assisted deprotonation pathway in OER. a** Atomic structures of the (110) surfaces of $IrO_2$–$O_v$ (left) and Mo–$IrO_2$–Ov. **b** The H atom adsorption energy on surface $O_{bri}$ sites in $IrO_2$–$O_v$ and Mo–$IrO_2$–$O_v$. Right panel is the schematic illustration. **c** Schematic illustrations of the catalyst structure after reconstruction (left) and a faster BOAD process upon Mo substitution (right). **d** The free-energy diagram of Mo–$IrO_2$–$O_v$ with different OER pathways. PDS is labeled by green star. Color code: Ir (green), Mo (purple), and O (red). The green and purple octahedra represent [$IrO_6$] and [$MoO_6$] octahedrons, respectively.

pathway, the PDS of Mo-doped $IrO_2$–$O_v$ shifts from $OOH_{top}^* \rightarrow O_2 + (H^+ + e^-)$ to $OO_{top}^* + OH_{bri}^* \rightarrow * + O_2 + (H^+ + e^-)$ with a decreased overpotential from 0.60 V to 0.43 V. (Fig. 7d, PDS is labeled by green star). Consequently, both $IrO_2$–$O_v$ and Mo-doped $IrO_2$–$O_v$ are prone to follow BOAD pathway, and the thermodynamic OER overpotential is decreased by 0.51 V with Mo substitution after surface reconstruction (Fig. 7d). To verify the feasibility of the chemical steps, the rate constant of the second $H_2O$ dissociation ($O_{top}^* + H_2O \rightarrow OOH_{top}^* + OH_{bri}^*$) was further discussed by the transition state theory (TST). According to the climbing image nudged elastic band (CINEB) calculation[67], its corresponding activation-free energy barrier is demonstrated as 0.64 eV (Supplementary Fig. S18), which is below the hard limit for a surmountable barrier (0.75 eV) at room temperature, indicating that the chemical steps in BOAD can process readily (Supplementary Fig. S21 and Supplementary Note 6)[68]. As depicted, adsorption of $O_{top}$ shows no noticeable change after Mo substitution, whereas the deprotonation process on $O_{bri}$ has been significantly facilitated due to the stronger Brønsted acidity of Mo$-O_{bri}$–Ir, in sharp contrast to the case in Ir$-O_{bri}$–Ir, where the H adsorption of $OH_{bri}$ is rather strong, resulting in a large overpotential. These indicate that introduction of Mo has strengthened the Brønsted acidity of $O_{bri}$, thus optimizing the proton transfer process in OER following BOAD pathway.

To verify the universality of Mo doping in regulating OER performance, we synthesized Mo-doped $IrO_2$ (Mo–$IrO_2$) and Mo-doped $RuO_2$ (Mo–$RuO_2$) via sol–gel method (see "Methods" for details) and further compare the electrochemical performance with their undoped counterparts. It is found that Mo also improves the performance of $IrO_2$ and $RuO_2$ systems in acidic OER (Supplementary Figs. S19 and S20 and Supplementary Notes 7 and 8).

## Discussion
In summary, using $Pr_3Ir_{1-x}Mo_xO_7$ as model, we have developed a facile electrochemical reconstruction strategy to construct highly active and self-terminated Ir–$O_{bri}$–Mo species for water oxidation in acidic electrolytes. The presence of high-valence Mo accelerates surface reconstruction due to the optimized Ir–O covalency and more prone dissolution of Pr. Meanwhile, excessive loss of lattice oxygen is effectively avoided benefitting from Mo-buffered charge compensation. Significantly, highly active Ir–$O_{bri}$–Mo species as strong Brønsted acid in surface reconstruction layers facilitate deprotonation of oxo intermediates following BOAD pathway, resulting in an overall activity improvement. This work proposes a facile strategy for constructing strong Brønsted acid sites in $IrO_x$ through directional surface reconstruction of iridates, demonstrating the perspective of targeted electrocatalyst fabrication under in situ realistic reaction conditions.

## Methods
### Materials synthesis
A conventional solid-state method was employed to synthesize $Pr_3Ir_{1-x}Mo_xO_7$ ($x$ = 0, 0.1, 0.2, 0.4). Powder precursors praseodymium (III) acetate hydrate (Macklin, 99.9%), iridium (IV) chloride hydrate (Macklin, Ir 48.0–55.0%), and molybdenum trioxide (Macklin, 99.95%) were used as raw materials. Briefly, stoichiometric raw materials were weighted and thoroughly ground in agate mortar. The mixed precursors were calcined in ambient air at 1400 °C for 5 h to obtain $Pr_3Ir_{1-x}Mo_xO_7$ ($x$ = 0, 0.1, 0.2, 0.4) which are denoted as $x$Mo-PIO throughout this work. Acid-treated samples denoted as $x$Mo-PIO-acid were obtained via soaking $x$Mo-PIO in 0.1 M $HClO_4$ for 3 h. The powder was separated from the solution by centrifugation, rinsed with milli-Q water, then dried for use.

As comparison, Mo-doped $IrO_2$ precursor was synthesized by sol–gel method and then calcined in air to obtain the oxide powder. For the synthesis of precursor, 0.025 g molybdenum(V) chloride (Macklin, 99.5%) was firstly dissolved in 4 mL ethylene glycol (J&K Scientific, 99%, extra pure), then 0.12 g iridium (IV) chloride hydrate (Macklin, Ir 48.0–55.0%) was added. After dissolution, 0.16 g citric acid (J&K Scientific, 99.5%, anhydrous, ACS reagent) which has been beforehand dissolved in 15 mL water and 0.1 mL ammonia solution (Aladdin, GR, 25–28%) was dropped into above solution. The resulting solution was stirred at 90 °C for 8 h, and then heated in an oven at

170 °C for 12 h to obtain a solid precursor. Then, this precursor was calcined in air at 550 °C for 6 h to acquire the targeted catalysts. Undoped $IrO_2$ was synthesized without adding molybdenum(V) chloride. Mo-doped $RuO_2$ was synthesized by the same method as Mo-doped $IrO_2$ except for choosing ruthenium (III) chloride (J&K Scientific, 99%, anhydrous) rather than iridium (IV) chloride hydrate. Undoped $RuO_2$ was synthesized without adding molybdenum(V) chloride following the same procedure as Mo–$RuO_2$.

### Materials characterizations

The powder X-ray diffraction (XRD) patterns was carried out with a Bruker D8 Focus operating at 40 kV and 40 mA equipped with a nickel-filtered Cu Kα radiation ($\lambda$ = 1.541 Å). Field-emission scanning electron microscopy (FE-SEM) images were obtained using a Hitachi S-4800 SEM. Transmission electron microscope (TEM) was conducted on JEM-F200 with a field-emission gun operating at 200 kV. X-ray photoelectron spectrum (XPS) data was collected by a PHI-1600 X-ray photoelectron spectroscope equipped with Al Kα radiation with the binding energy was calibrated by C 1s peak at 284.8 eV. Brunauer–Emmett–Teller (BET) surface areas of powder samples were determined using $N_2$ adsorption/desorption isotherms recorded on a Micromeritics TriStar 3000 instrument at 77 K. Raman spectra were collected on a Horiba LabRAM HR Evolution Raman microscope using a 532 nm laser excitation with a 50x objective lens. Each Raman spectrum was acquired over a collection time of 15 s and is the average of three measurements. The in situ Raman was performed during the electrochemical test by using a constant potential mode, where the applied potentials were increased step by step from 0.70 to 1.60 V (versus RHE).

### Electrochemical testing

Electrochemical measurements were conducted using an IVIUMSTAT (Ivium Technologies BV, Netherlands) workstation and a Pine rotating disk electrode (RDE) apparatus in a typical three-electrode setup with 0.1 M $HClO_4$ solution as electrolyte, a clean Pt gauze as the counter electrode and a saturated calomel electrode (SCE) as the reference electrode (Supplementary Fig. S21). The reference electrode SCE was calibrated according to the method reported by Boettcher and coworkers[69]. In the three-electrode system, the reference electrode is a SCE, two Pt plates were used as the counter electrode and the working electrode, respectively. Before the calibration, the electrolyte 0.1 M $HClO_4$ should be saturated with $H_2$. During the calibration (LSV test at 2 mV s$^{-1}$), hydrogen was bubbled over the working electrode. The correction potential is the potential of zero net current. (Supplementary Fig. S22). The as-measured potentials (versus SCE) were calibrated by E (vs. RHE) = E (vs. SCE) + 0.273 V − $iR_s$. $R_s$ is solution resistance determined by electrochemical impendence spectra (EIS) test. The non-$iR$ corrected LSV curves are given in Supplementary Fig. S23. The EIS was tested in a range of 0.01–100 kHz at 1.3 V (versus SCE). A glassy carbon electrode with a diameter of 5 mm covered by a thin catalyst film was used as the working electrode. Typically, 5 mg catalyst and 1 mg Vulcan carbon were suspended in 970 μL isopropanol-water solution with 30 μL Nafion to form a homogeneous ink assisted by ultrasound for 3 h. Then, 10 μL of ink was dropped onto the glassy carbon (0.196 cm$^{-2}$) and dried overnight he (mass loading: 0.25 mg cm$_{geo}^{-2}$). Linear sweep voltammetry (LSV) was performed at a scan rate of 5 mV s$^{-1}$ to eliminate any contribution from the capacitive effect and at a rotation speed of 1600 rpm to dissipate the generated oxygen bubbles to obtain OER kinetic currents. The potentials were $iR$ corrected to compensate for solution resistance. Double layered capacitances ($C_{dl}$) were assessed by cyclic voltammetry (CV) to estimate the electrochemically active surface area (ECSA) with scan rate from 20 to 120 mV s$^{-1}$ at potential window of 0.865–0.975 V (versus SCE). A specific capacitance of 40 μFcm$^{-2}$ was applied.

### Stability number (S-number) calculation

The S-number was calculated by the following equation as previously reported:[70]

$$S - number = \frac{n_{O2}}{n_{Ir(dissolved)}} \tag{1}$$

where $n_{O2}$ and $n_{Ir\ (dissolved)}$ refer to the total amount of evolved oxygen (calculated from total charge) during the chronopotentiometry test and the amount of dissolved Ir extracted from ICP-MS results, respectively.

### Computational methods

The spin-polarized calculations within the density functional theory (DFT) framework were carried out by the Vienna ab initio simulation package (VASP)[71]. The interactions between core and electrons were represented by the projector-augmented wave (PAW) method and the generalized gradient approximation (GGA) with the Perdew–Burke–Ernzerhof (PBE) exchange-correlation functional[72,73]. And the valence configuration of the PAW potentials for Pr, Ir, Mo, and O are 11, 9, 6, and 6, respectively. In addition, the Hubbard-corrected DFT functional, PBE + U was utilized[74]. An effective U value was adopted at 4.38 eV for Mo. $Pr_3rO_7$ crystal with *Cmcm* space group was selected, and the optimal lattice parameter is a = 7.55 Å, b = 11.07 Å, c = 7.63 Å. For Mo-doped $Pr_3rO_7$, the Ir site is substituted with one Mo atom, and the corresponding lattice parameter have a little variation with a = 7.56 Å, b = 11.08 Å, c = −7.67 Å. The $IrO_2$(110)-$O_v$ surfaces were modeled with a slab of three atomic layers in which the bottom one layer was frozen, and a vacuum layer of about 15 Å along the z-axis was built. An O vacancy ($O_v$) was introduced due to the results of XPS. For Mo-doped $IrO_2$(110)-$O_v$, one Mo atom near the $O_v$ in the first layers of $IrO_2$(110)−$O_v$ was replaced by a Mo atom (Supplementary Data). For (Mo−) PIO and (Mo−) $IrO_2$(110)-$O_v$, a cutoff energy of 520, and 450 eV was employed for the plane-wave basis set, respectively. The Brillouin-zone integrations were performed using a (3 × 2 × 3) and a (4 × 2 × 1) Monkhorst–Pack mesh, respectively. The iterative process considered was convergences, when the force on the atom was <0.05 eV Å$^{-1}$ and the energy change was <10$^{-4}$ eV per atom (see Supplementary Methods for details). The Gibbs free energies at 298.15 K and 1 atm were calculated by:

$$G = H-TS = E_{DFT} + E_{ZPE} + \int_0^{298.15\,K} C_V dT - TS \tag{2}$$

where $E_{DFT}$ is the total energy obtained from DFT optimization, $E_{ZPE}$ is the zero-point vibrational energy using the harmonic approximation[75], $C_V$ is the heat capacity, T is the kelvin temperature, and S is the entropy. The entropies of gas molecules were taken from NIST database. The free energy of liquid water was calculated as an ideal gas at 3534 Pa, which corresponds to the vapor pressure of water[76]. The computational hydrogen electrode (CHE) model was used to calculate the free energy of electrocatalytic OER[77]. Climbing image nudged elastic band (CINEB) was used for calculating the free-energy barrier ($\Delta G_{TS}$) of transition state[67].

To evaluate the stability of the metals and O atoms, the formation energies ($G_{f-vacancy}$) of vacancies were proposed by DFT calculation and followed the equation:

$$G_{f-vacancy} = G_{vacancy} + G_{M/O} - G_{perfect} \tag{3}$$

where $G_{perfect}$ represents the energy of the perfect compounds, $G_{vacancy}$ is the energy of compounds with a M/O vacancy, $G_{M/O}$ is the energy of a M/O atom.

To map the Pourbaix diagram of Mo–$IrO_2$–$O_v$, the formation energies ($G_i^0$) at standard state (298.15 K, 1.0 bar) of elements were

used as the references to compute the Gibbs energies ($G_i^f$) of the Mo–$IrO_2$–$O_v$ and its derived compounds. The $G_{solid}^0$ of solids elements was computed by DFT calculation and obtained by the equation:

$$G_{solid}^0 = G_{Mo_xIr_yO_{z-x}}G_{Mo-y}G_{Ir-z}GO \qquad (4)$$

in which the $G_{Mo_xIr_yO_z}$ represents the formation energy of the $Mo_xIr_yO_z$ (Mo–$IrO_2$–$O_v$, $IrO_2$–$O_v$, bulk Mo, $MoO_2$, $MoO_3$), and the $G_{Mo}$, $G_{Ir}$ represent the formation energies of single Mo, Ir atom, which obtained from bulk Mo and Ir, respectively. The Gibbs formation energy of O is −4.57 eV obtained from ref. 78. In addition, the $G_{ion}^0$ of derived ions ($Mo^{3+}$, $[MoO_4]^{2-}$) were obtained from the experimental database and then corrected by a formation of Gibbs free-energy difference between the calculated reference solid ($MoO_2$) and its experimental respective value[78]. The corresponding equation is:

$$G_{ion}^0 = G_{ion}^{Exp} + \left[ G_{MoO2}^{DFT} - G_{MoO2}^{Exp} \right] \qquad (5)$$

in which the $G_{ion}^{Exp}$ is the formation energy of ions obtained from experiments. The $G_{MoO2}^{DFT}$, $G_{MoO2}^{Exp}$ denotes the DFT-calculation and experimental formation energies of $MoO_2$, respectively. The correction has been demonstrated can combine the DFT calculated solids with experimental values of arbitrary ions, thus reproducing the dissolution of the solids. The x, y, z are the stoichiometric coefficients of Mo, Ir, O, respectively. The Gibbs free energy of each species (i = solid, ion) can be expressed as:

$$G_i^f = G_i^0 + 0.0591\log c_i - n_O\mu_{H2O} + pH(n_H - 2n_O) + \varphi(2n_O - n_H + q_i) \qquad (6)$$

In Eq. (6), $c_i$ is the concentration of species i, which is 1 for solids and $10^{-6}$ for ions. $\mu_{H2O}$ is the formation of water, $n_O$ and $n_H$ are the containing oxygen and hydrogen atoms numbers in the species, $\varphi$ is the electric potential, while $q_i$ is the charge number of considered species. With this equation, we can describe the Gibbs free energy of the solid phases as a function of pH and applied potential $\varphi$. We implemented the Pourbaix diagram by pymatgen to study the stability of Mo–$IrO_2$–$O_v$, $Mo^{3+}$ + $IrO_2$–$O_v$, Mo + $IrO_2$–$O_v$, $MoO_2$ + $IrO_2$–$O_v$, $MoO_3$ + $IrO_2$–$O_v$, and $(MoO_4)^{2-}$ + $IrO_2$-$O_v$[78–80]. The result shows that the $Mo^{3+}$ + $IrO_2$–$O_v$ and $MoO_2$ + $IrO_2$–$O_v$ is unstable, while the other four compounds exhibit favorable stability. The stability of Mo–$IrO_2$–$O_v$ is set as the reference material, which is 0 eV/atom.

## Data availability
The data that support the findings of this study are provided in the Supplementary Information and Source Data file, are available from the corresponding authors upon reasonable request. Source data are provided with this paper.

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

## Acknowledgements

The authors appreciate the support from the National Key R&D Program of China (2020YFA0710000), the National Natural Science Foundation of China (22278307, 22008170, 22222808, 22121004), and the Haihe Laboratory of Sustainable Chemical Transformations.

## Author contributions

J.-J.Z. and Z.-F.H. designed the studies. S.C. synthesized the catalysts, performed the catalytic tests, and wrote the paper. S.Z. performed the density functional theory calculations. S.C., S.Z., L.G., L.P., C.S., X.Z., and G.Y. conducted and analyzed structural characterizations. J.-J.Z., Z.-F.H., and G.Y. revised and polished this paper. All authors discussed the results and commented on the manuscript.

## Competing interests

The authors declare no competing interests
