## [Peer Review File · Nature Communications]

REVIEWER COMMENTS

Reviewer #1 (Remarks to the Author):

The authors present a combined experiment/theory study of the structural evolution and catalytic performance of Mo-doped Pr₃IrO₇. Their results show that Mo-doping leads to a surface restructuring involving Pr leaching. The resulting surface exhibits highly active species formed by Ir and Mo bridged by an oxygen atom. The authors argue that this surface enables a bridging-oxygen-assisted-deprotonation pathway with a reduced overpotential.

While the experiments are extensive and seem - as far as a theoretician can judge - to be carefully conducted, the supporting DFT calculations leave to desire. Also, the discussion of results is not always sufficiently exhaustive to be accessible to a non-specialist reader, which, in my eyes, is a requirement for publication in a general-audience journal such as Nature Communications. Therefore, I cannot recommend publication of this manuscript. Following is a detailed list of points of criticism.

1) The motivation in the introduction presently does not make complete sense. For example, it is not clear from the cited literature why a multi-valent dopant such as Mo can prevent Ir dissolution. Also, why is Mo chosen here and would other transition metals work as well or maybe even better?

2) The language will need major revision. A large part of the manuscript is really difficult to understand and, as such, will presently not appeal to a wide readership.

3) Computational methods: i) the authors use plain PBE, which gets redox energetics of Mo oxides very wrong. The materials project recommends using a Hubbard U correction of 4.38 eV on Mo. The authors need to carefully evaluate what is the effect of U on their results and, in case a large effect is observed, recalculate the main results with a more appropriate setup than plain PBE. ii) the valence configuration of the PAW potentials needs to be given to allow the reproduction of the results. iii) no information on the computational setup of the surface calculations is given.

4) Charge density differences are always given with respect to something. Therefore it is very unclear what figure 1c represents and how it should be interpreted.

5) The authors use bulk calculations in presence of a Mo dopant and/or oxygen vacancies to explain stability and bonding. This is a huge approximation, as defect formation at surfaces and in the bulk can

deviate significantly. Since the authors anyway study surfaces for the OER cycles, the same sampling of defects should be performed at the surface.

6) The authors invoke shifts of the band centers to explain changes in bonding. While I can follow for the O 2p bands, the discussion related to the Ir 5d band is not clear to me.

7) The schematic in Fig. 5e is impossible to understand. Where is the surface and what does the reconstruction look like? The same is true for many of the figures, like figure 7a or S12, where the viewing direction is unclear.

8) In Fig. 8 the PDS should be highlighted to ease the discussion.

9) When discussing the different OER pathways, the authors assume that chemical steps associated with the BOAD pathway are readily overcome. Is this really justified for steps of more than 1 eV? What temperatures would be required to enable these processes on a reasonable time scale?

Reviewer #2 (Remarks to the Author):

In this work, the authors have demonstrated $\text{Pr}_3\text{Ir}_{1-x}\text{Mo}_x\text{O}_7$ with high-valence Mo modulated to be a good catalyst toward acidic OER. The accelerated surface reconstruction with Mo doping also leads to buffered charge compensation, which prevents the fierce Ir dissolution and excessive lattice oxygen loss. The results are interesting and supported by comprehensive theoretical predictions and solid experimental evidences. However, there are still some issues to be addressed before being publishing in Nature Communication. Please find the comments below.

1. The KIE analysis in Fig.6d is not rigorous at the current state. First, only the H_2O was replaced by the D_2O , however, the HClO_4 was used in both electrolyte; the H is not completely substituted by D. Secondly, the potential scale was normalized on the RHE scale, rather the overpotential scale. Actually, the pD/pH, and the equilibrium potential for $\text{D}_2\text{O}/\text{O}_2$ and $\text{H}_2\text{O}/\text{O}_2$ are all different. Please check the literature and do a more rational analysis.

2. The authors selected weberite type Pr_3IrO_7 material systems for acid OER research to study the effect of Mo doping on catalyst reconstruction and active substances. As revealed by the authors, the active substance is Ir-O-Mo motif. Then this brings up a curiosity about whether Mo doping also has a positive effect in the IrO_2 system. The universality of this Mo doping should be verified in other systems.

3. In this work, OER performance was assessed in 0.1M HClO₄ electrolyte, which is different from those in literature (usually pH ≈ 0). It would be helpful to provide the OER performance tested in 1M HClO₄ or 0.5M H₂SO₄, and then compared to other acidic OER catalysts.

4. The stability measurement is conducted at the geometric current density of 5 mA cm⁻², which is quite low compared to the results in the literature. The measurement on higher current density should be provided to make a fair comparison. Besides, the Ir leaching should be also investigated. The stability number should be calculated and compared to other Ir-based catalysts in the literature.

5. In introduction, authors' claimed that "the mass content of iridium in Ln₃IrO₇ compounds (26.4% in Pr₃IrO₇) is obviously lower than that in IrO₂ (85.7%), perovskite and pyrochlore structures (58.6% in SrIrO₃ and 49.4% in Pr₂Ir₂O₇). Consequently, if properly tuned, directional surface reconstruction with enhanced stability and further decreased iridium consumption of Ln₃IrO₇ will be anticipated." It is not rigorous, since in the PEM electrolytic water system, we are more concerned about the loading amount of precious metals on the membrane electrode. It is more convincing to compare the mass activity (normalized to Ir mass) of the as-prepared catalysts herein to other Ir-based catalysts.

6. The reference citation in the following sections is insufficient.

☒ On line 179-187, the discussions on Figure 2e.

☒ On line 213-218, "Two successive oxidation peaks..." and "...a deprotonation process of active oxygen intermediates regulated by Mo."

☒ On line 228-229, "...probably modified OER pathway..."

☒ In the part of discussing about Mo-buffered charge compensation during reconstruction, there is a lack of extensive reference here.

7. Please label the Raman peaks in Fig. 6e so it is easier to understand the structure evolution.

8. In figure 5b and 5c, the peak for Ir<V is located at the lower energy compared to that for IrIII, indicating that the valence is below +3. Please provide more details on this analysis.

9. In figure 2d, please double check the scale in the x axis.

Reviewer #3 (Remarks to the Author):

The manuscript „Reconstructed Ir–Obri–Mo species with strong Brønsted acidity for acidic water oxidation” by Chen et al describes the synthesis and investigation of Pr₃Ir_{1-x}Mo_xO₇ and the resulting Ir–Obri–Mo active species by different methods, including DFT, XPS, Raman and electrochemical measurements. Currently, green hydrogen plays a crucial role in the global energy discussion, and with this comes also a special importance of iridium based OER catalysts. The here presented results show new insights and solution in how to improve the stability and activity of the iridium based catalysts. It is shown how the addition of new elements into Iridium oxide (namely Pr and Mo) can improve the

performance of the catalyst, and also that the conversion during the OER has to be considered. The manuscript gives deep insights into the appearing processes and the properties of the resulting species. The graphs are presented in a very nice and clear manner and the discussion in the text is easy to understand and to follow. Overall, it is around the most easiest to understand and most clear structured paper I have read within the last few months.

The combination of the theoretical calculations and the electrochemical testing, as well as the characterization of the catalyst before and after the catalyst testing, makes this a nice manuscript with high interest to the research community.

I therefore suggest this manuscript to be accepted for publications after addressing the following minor points:

- It is not totally clear to me how the authors choose Pr₃IrO₇ as a system to do DFT calculations on. Therefore, some sentences should be added at the beginning to make the line of thought of the authors more clear.
- Likewise, a short discussion and outlook should be added at the end to put this research work into the wider perspective

Best regards,

Steffen Czoska

We thank the reviewers for the detailed and constructive comments in improving the quality of this manuscript. Provided below is our detailed point-to-point response to each question. The changes in the manuscript have been highlighted and listed below.

Reviewer #1

The authors present a combined experiment/theory study of the structural evolution and catalytic performance of Mo-doped Pr_3IrO_7 . Their results show that Mo-doping leads to a surface restructuring involving Pr leaching. The resulting surface exhibits highly active species formed by Ir and Mo bridged by an oxygen atom. The authors argue that this surface enables a bridging-oxygen-assisted-deprotonation pathway with a reduced overpotential.

While the experiments are extensive and seem - as far as a theoretician can judge - to be carefully conducted, the supporting DFT calculations leave to desire. Also, the discussion of results is not always sufficiently exhaustive to be accessible to a non-specialist reader, which, in my eyes, is a requirement for publication in a general-audience journal such as Nature Communications. Therefore, I cannot recommend publication of this manuscript.

Following is a detailed list of points of criticism.

Comment 1:

The motivation in the introduction presently does not make complete sense. For example, it is not clear from the cited literature why a multi-valent dopant such as Mo can prevent Ir dissolution. Also, why is Mo chosen here and would other transition metals work as well or maybe even better?

Response 1:

We thank the reviewer for the comment. Firstly, preferential OER mechanisms depend heavily on the electronic structure of electrocatalysts. Studies have suggested that electrocatalysts with high metal-oxygen covalency are expected to follow lattice oxygen-mediated mechanism (LOM) because of the strong oxygen character of the band near the Fermi level (*Nat. Energy* **2**, 16189 (2016); *Nat. Commun.* **7**, 11053 (2016)). Consequently, undoped Pr_3IrO_7 is thought to suffer lattice oxygen activation under oxidation potential because of the large O 2p contribution around Fermi level (Figure 1d). The surface is no longer thermodynamically stable upon lattice oxygen activation. The subsequent lattice oxygen regeneration (vacancies refilled by H_2O) is generally kinetically sluggish, resulting in accumulation of oxygen vacancies (*Nat. Mater.* **16**, 925 (2017); *Sci. Adv.* **7**, eabc7323 (2021)). To achieve charge neutrality, metal cations (both Ir and Pr) are prone to dissolve under this condition for charge compensation. So here comes the strategy that multi-valent Mo can realize charge compensation because of the increase in oxidation state during surface reconstructions. Secondly, doping elements are usually completely leached during surface reconstruction in acidic media (*Nat. Commun.* **10**, 572

(2019); *Sci. Adv.* **7**, eabk1788 (2021)). Indeed, the leached elements help with regulating reconstruction process, but they fail to work within the reconstructed layers. As a result, acid-stable doping element will be a priority. According to the *acid-stable periodic table* built by Nørskov's group (*ACS Energy Lett.* **5**, 2905 (2020)), Sb/Ti/Sn/Ge/Mo/W-based oxides have preference for being stable in strong acids, among which Mo and W are multi-valent. Meanwhile, taking the crystal structure of Pr_3MO_7 (M can be Ir, Ru, Os, Mo, Ta) into consideration, Mo is capable of occupying the Ir site to form similar crystal structure. Besides, Mo is earth abundant and the cheapest compared with noble metals (Ir, Ru and Os) and Ta. As a result, Mo is chosen finally.

Modification 1:

1). Page 2, line 43-45, two references are added as follows.

Iridates with weakly corner-shared IrO_6 configuration exhibit high initial lattice oxygen reactivity whereas dramatically deteriorate soon because of its kinetically sluggish regeneration, leading to drastic cations leach-out and continuous Ir loss.^{9,10}

9. Wan G, *et al.* Amorphization mechanism of SrIrO_3 electrocatalyst: How oxygen redox initiates ionic diffusion and structural reorganization. *Sci. Adv.* **7**, eabc7323 (2021).

10. Song CW, Lim J, Bae HB, Chung S-Y. Discovery of crystal structure–stability correlation in iridates for oxygen evolution electrocatalysis in acid. *Energy Environ. Sci.* **13**, 4178-4188 (2020).

2). Page 2, line 62-65, one sentence is revised as follows.

Considering the crystal structure of Ln_3IrO_7 , the multi-valent nature of Mo, and the modulation of Brønsted acidity by Mo and the acidic-stability of molybdenum oxide,^{4, 5, 21} it is expected that substitution of Ir in Ln_3IrO_7 with Mo can realize controllable surface reconstruction towards desired $\text{Ir}-\text{O}^{(\text{II}-\delta)^-}-\text{Mo}$ active species with reduced Ir consumption and optimized electronic configuration.

Comment 2:

The language will need major revision. A large part of the manuscript is really difficult to understand and, as such, will presently not appeal to a wide readership.

Response and Modification 2:

We thank the reviewer for the comment. We are sorry for the obscure expressions in the manuscript. As suggested by the reviewer, to make the manuscript appeal to general-audience, we have revised the language and simplified the expressions.

Comment 3:

Computational methods: i) the authors use plain PBE, which gets redox energetics of Mo oxides very wrong. The materials project recommends using a Hubbard U correction of 4.38 eV on Mo. The

authors need to carefully evaluate what is the effect of U on their results and, in case a large effect is observed, recalculate the main results with a more appropriate setup than plain PBE. ii) the valence configuration of the PAW potentials needs to be given to allow the reproduction of the results. iii) no information on the computational setup of the surface calculations is given.

Response 3:

We thank the reviewer for the valuable and perceptive comment.

i) As suggested, we have recalculated the Mo-PIO and Mo-IrO₂-O_v systems by PBE + U to evaluate the effect of U on Mo. In general, the trends of vacancy formation energy, reaction free energy difference and electronic distribution have kept consistent with the previous PBE calculations, so the same conclusions can be drawn. In detail, there are some differences in the energy values after a Hubbard U correction of 4.38 eV on Mo atoms. **Firstly**, all the formation energies (E_f) of different Mo-substitution sites have increased, but the Ir-site substitution still exhibits the most favorable energy (0.25 eV) (Table S1). **Secondly**, the geometric properties of Mo-PIO have a little change (Figure S2). **Thirdly**, the electronic properties of Mo-PIO, such as charge density and Bader charge, have a little change, which also demonstrate the charge transfer from Mo to Ir. Specifically, the Bader charge of Mo and O has changed from -2.32 |e| to -2.38 |e| and from 32.80 |e| to 32.86 |e|, respectively. The Bader charge of Pr and Ir has not changed, indicating the obtained charge of Ir from Mo (0.16 |e|) is not affected by the PBE + U calculations (Figure 1c, Table S2). **Fourthly**, the trends of projected density of states (PDOS) are consistent with the PBE results. The charge-transfer energy of Mo-PIO decreases from 0.12 to 0.07, which lower than that of PIO. The O 2p band center of O₍₁₎, O₍₂₎, O₍₃₎ also have a little change, but have not affected the original conclusion. (Figure 1d, Table S3). **Fifthly**, the vacancy formation energies in Table S4, Table S5 and Table S6 have some change, but the related conclusions are not influenced. **Sixthly**, the hydrogen adsorption energy (E_{ads}) on Mo-O_{bri}-Ir in Mo-doped IrO₂-O_v have changed from -0.65 eV to -0.78 eV (Figure 7b). **Seventhly**, the Hubbard U correction on Mo atom increases the adsorption strength of OH_{top}* and OH_{bri}* compared with PBE calculations. On account of the variation of the adsorption energy, the overpotential of Mo-doped IrO₂-O_v have increased from 0.58 eV to 0.60 eV for AEM, and that for BOAD have changed from 0.33 eV to 0.43 eV. However, the BOAD still exhibits favorable performance than traditional AEM, and the Mo substitution facilitates the OER process by decreasing the OER overpotential of 0.51 eV (Figure 7d).

ii) We are sorry for the negligence of the valence configuration of the PAW potentials. The valence of Ir, Mo and O are 9, 6 and 6, respectively. Specially, the valence of Pr is 11, and the 4f electrons of Pr are treated as core states (**Computational Methods**).

iii) We have added the information computational setup of the surface calculations. In addition, the surface and bulk calculations share similar methods, except for the cut-off energy and the Brillouin zone integrations, which have been highlighted in **Computational Methods** in the revised manuscript.

Modification 3:

1). Page 3, line 87-89, one sentence is revised as follows.

As seen, the formation energy is energy favorable for Ir-site substitution (0.25 eV) than that for Pr-site substitutions (7.94 eV for Pr₍₁₎, 9.57 eV for Pr₍₂₎), demonstrating that Mo atoms are preferred to occupy six-coordinated Ir sites (Figure S1 and Tables S1, Supporting Information).

2). Page 5, line 121-124, one sentence is revised as follows.

As shown in Table S5, O₍₁₎ vacancies are inclined to form in PIO, while after Mo substitution, vacancy formation energies have decreased for O₍₃₎ (from 2.83 eV to 2.66 eV) and O₍₁₎ in Mo-O₍₁₎ bonds (from 2.73 eV to 2.64 eV).

3). Page 5, line 124-128, one sentence is revised as follows.

As seen, Pr vacancy formation energies further decrease when oxygen atoms are removed from the lattice (9.58 eV for pristine samples compared with 8.50 eV and 9.37 eV after formation of O₍₁₎ and O₍₃₎ vacancies in Mo-doped Pr₃IrO₇), indicating probable surface reconstruction process mediated by activation of lattice oxygen coupled metal dissolution (Table S6).

4). Page 15, line 396-398, one sentence is revised as follows.

Ir-O_{bri}-Ir shows fairly strong H adsorption with an E_{ads} of -1.24 eV, whereas the adsorption strength is significantly weakened on Mo-O_{bri}-Ir with an E_{ads} of -0.78 eV (Figure 7b).

5). Page 16, line 421-426, two sentences are revised as follows.

As for Mo-doped IrO₂-O_v, with the introduction of BOAD pathway, the PDS of Mo-doped IrO₂-O_v shifts from OOH_{top}* → O₂ + (H⁺ + e⁻) to OO_{top}* + OH_{bri}* → * + O₂ + (H⁺ + e⁻) with a decreased overpotential from 0.60 V to 0.43 V. Consequently, both IrO₂-O_v and Mo-doped IrO₂-O_v are prone to follow BOAD pathway, and the thermodynamic OER overpotential is decreased by 0.51 V with Mo substitution after surface reconstruction (Figure 7d).

6). Figure 1c-1e are revised as follows.

Figure 1. DFT calculations on Mo-redirected surface reconstruction of Pr_3IrO_7 . (a) Crystal structures of Weberite type Pr_3IrO_7 . (b) Lattice distortion induced by Mo substitution. (c) Charge density difference for Pr_3IrO_7 (left) and $\text{Mo-Pr}_3\text{IrO}_7$ (right). Color code: Pr (blue), Ir (green), Mo (purple), and O (red). Red and green shadows represent charge accumulation and depletion, respectively. (d) PDOS of Ir 5d, Mo 3d and O 2p orbitals for Pr_3IrO_7 (top) and $\text{Mo-Pr}_3\text{IrO}_7$ (bottom). (e) PDOS of different kinds of lattice oxygen. (f) Schematic illustration for the accelerated surface reconstruction upon Mo substitution. (g) Pourbaix diagram of $\text{Mo-IrO}_2\text{-Ov}$.

7). Figure 7b and 7d are revised as follows.

Figure 7. Bridging-oxygen-assisted deprotonation pathway in OER. (a) Atomic structures of the (110) surfaces of IrO₂-O_v (left) and Mo-IrO₂-O_v. (b) The H atom adsorption energy on surface O_{br} sites in IrO₂-O_v and Mo-IrO₂-O_v. Right panel is the schematic illustration. (c) Schematic illustrations of the catalyst structure after reconstruction (left) and a faster BOAD process upon Mo substitution (right). (d) The free energy diagram of Mo-IrO₂-O_v with different OER pathways. PDS is labeled by green star. Color code: Ir (green), Mo (purple), and O (red). The green and purple octahedra represent [IrO₆] and [MoO₆] octahedrons, respectively.

8). **Supporting Information**, two sentences are added in **Computational Methods**.

And the valence configuration of the PAW potentials for Pr, Ir, Mo and O are 11, 9, 6 and 6, respectively. In addition, the Hubbard-corrected DFT functional, PBE + U was utilized.⁵ An effective U value was adopted at 4.38 eV for Mo.

The IrO₂(110)-O_v surfaces were modeled with a slab of three atomic layers in which the bottom one layer was frozen, and a vacuum layer of about 15 Å along the z-axis was built. An O vacancy (O_v) was introduced due to the results of XPS. For Mo-doped IrO₂(110)-O_v, one Mo atom near the O_v in the first layers of IrO₂(110)-O_v was replaced by a Mo atom.

5. Anisimov VI, Zaanen J, Andersen OK. Band theory and Mott insulators: Hubbard U instead of Stoner I. *Phys. Rev. B* **44**, 943-954 (1991).

9). **Supporting Information, Computational Methods**, parts of two sentences have been highlighted. For (Mo-) PIO and (Mo-) IrO₂(110)-O_v, a cut-off energy of 520, and 450 eV was employed for the plane-wave basis set, respectively. The Brillouin-zone integrations were performed using a (3×2×3) and a (4×2×1) Monkhorst-Pack mesh, respectively.

10). **Supporting Information, Figure S2** is revised as follows.

Figure S2. (a), (b) Corresponding bond length and bond angle of Pr_3IrO_7 (a) and Mo-doped Pr_3IrO_7 (b)

11). Supporting Information, Table S1 is revised as follows.

Table S1. Formation energy of different Mo-substitution sites

Lattice site	Formation energy (E_f) / eV
Ir	0.25
Pr ₍₁₎	7.94
Pr ₍₂₎	9.57

12). Supporting Information, Table S2 is revised as follows.

Table S2. Bader charge analysis

	Pr / e	Ir / e	Mo / e	O / e
Pr_3IrO_7	-25.35	-7.02	null	32.37
Mo- Pr_3IrO_7	-25.38	-5.10	-2.38	32.86

13). Supporting Information, Table S3 is revised as follows.

Table S3. Band center and charge transfer energy determined by DOS

	$\epsilon_{\text{O-p}}$ / eV	$\epsilon_{\text{Ir-d}}$ / eV	$ \epsilon_{\text{Ir-d}} - \epsilon_{\text{O-p}} $ / eV
Pr_3IrO_7	-2.13	-2.45	0.32
Mo- Pr_3IrO_7	-2.21	-2.09	0.12

14). Supporting Information, Table S4 is revised as follows.

Table S4. Vacancy formation energy of Pr in Pr_3IrO_7 and Mo- Pr_3IrO_7

	Pr ₍₁₎ / eV	Pr ₍₂₎ / eV
Pr ₃ IrO ₇	10.34	10.16
Mo-Pr ₃ IrO ₇	9.92	9.58
Δ	-0.42	-0.58

15). **Supporting Information, Table S5** is revised as follows.

Table S5. Vacancy formation energy of O in Pr₃IrO₇ and Mo-Pr₃IrO₇

	Ir-O ₍₁₎ / eV	Mo-O ₍₁₎ / eV	O ₍₂₎ / eV	O ₍₃₎ / eV
Pr ₃ IrO ₇	2.73	null	2.93	2.83
Mo- Pr ₃ IrO ₇	3.02	2.64	3.23	2.66

Ir-O₍₁₎ and Mo-O₍₁₎ refer to O atoms bonded with Ir and Mo respectively.

16). **Supporting Information, Table S6** is revised as follows.

Table S6. Pr₍₂₎ vacancy formation energy in Pr₃IrO₇ and Mo-Pr₃IrO₇ with/without lattice oxygen vacancies

	E _{f-Pr(2)} / eV		
	pristine	O ₍₁₎ -vacancy	O ₍₃₎ -vacancy
Pr ₃ IrO ₇	10.16	8.79	9.71
Mo- Pr ₃ IrO ₇	9.58	8.50	9.37

Comment 4:

Charge density differences are always given with respect to something. Therefore, it is very unclear what figure 1c represents and how it should be interpreted.

Response 4:

We thank the reviewer for the comment. We are sorry for the negligence of detail information about the charge density differences. Detailed description about the charge density differences has been added. And the interpretation for Figure 1c has been highlighted in the revised manuscript.

Modification 4:

1). **Page 3, line 92-93**, one sentence is highlighted as follows.

Charge density difference also verifies the charge redistribution upon Mo substitution (Figure 1c).

2). **Page 3, line 93-94**, one sentence is added as follows.

The difference was implemented by subtracting the partial charge density of the Ir, Mo, Pr, and O atoms from the whole charge density of the (Mo-) Pr₃IrO₇.

3). Page 3, line 94-98, two sentences are highlighted as follows.

An obvious charge accumulation (red shadows) is observed around Ir sites accompanied with charge depletion (green shadows) around Mo sites. The Bader charge analysis further quantifies that the Ir atoms of Mo-PIO show the lower (0.16 |e|) charge depletion than that of PIO, indicating the electrons are transferred from Mo to Ir (Table S2).

Comment 5:

The authors use bulk calculations in presence of a Mo dopant and/or oxygen vacancies to explain stability and bonding. This is a huge approximation, as defect formation at surfaces and in the bulk can deviate significantly. Since the authors anyway study surfaces for the OER cycles, the same sampling of defects should be performed at the surface.

Response and Modification 5:

We thank the reviewer for the valuable comment. In previous studies, the formation energy of oxygen vacancy in bulk was used for estimating the possibility of surface reconstruction (*Adv. Mater.* **30**, 1802912 (2018). *Nat. Mater.* **16**, 925-931 (2017); *Nat. Commun.* **4**, 2439 (2013)). Besides, Calle-Vallejo *et al.* have demonstrated that there is a scaling relationship between bulk and surface properties for the metal oxides (*ACS Catal.* **5**, 869-873 (2015)). Consequently, it is reasonable to use the bulk properties to describe trends on surface reactivity for reconstruction. In addition, the trends of the surface reconstruction obtained by the formation energies of defects is consistent with the conclusions drawn from experiments. That is, electrochemical activation of outmost surface lattice oxygen and the initial dissolution of Pr are responsible for surface reconstruction. More fierce leaching of Pr and promoted lattice oxygen reactivity induced by Mo substitution contribute to deeper reconstruction with more exposure of catalytically active sites. Besides, charge compensation to Pr dissolution via increase of Mo oxidation state during reconstruction effectively avoids excessive loss of lattice oxygen and enables opportune reconstruction into self-terminated Ir–O_{bri}–Mo species.

It is true that the chemical reaction occurs on catalytic surface. Therefore, we further conducted DFT calculations based on the most thermodynamically stable IrO₂ (110) surface for elucidating the role of Mo for OER in reconstructed layers (Figure 7). As seen, the introduction of Mo has strengthened the Brønsted acidity of O_{bri} in Ir–O_{bri}–Mo species, thus optimizing the proton transfer process in OER following BOAD pathway. The potential determining step is the elemental step of OO_{top}* + OH_{bri}* → * + O₂ + (H⁺ + e⁻) with a lower overpotential of 0.43 V than that of Ir–O_{bri}–Ir species (0.94 V).

Comment 6:

The authors invoke shifts of the band centers to explain changes in bonding. While I can follow for the O 2p bands, the discussion related to the Ir 5d band is not clear to me.

Response 6

We thank the reviewer for the comment. The difference between O 2p-band center and Ir 5d-band center denotes the charge-transfer energy, which we have given in Table S3 and Figure 1d. Furthermore, Ir 5d-band center increases by 0.36 eV after Mo substitution compared with the slightly decreased O 2p-band center (0.08 eV). Consequently, the enlarged Ir–O covalency mainly originates from the increase of Ir 5d-band center. We have highlighted the related parts in the revised manuscript.

Modification 6:

1). Page 4, line 107-112, two sentences are highlighted as follows.

Figure 1d shows the calculated projected density of states (PDOS) of bulk PIO and Mo-PIO. The O 2p-band center and Ir 5d-band center are displayed by integrating the PDOS and the difference between them denotes the charge-transfer energy. As depicted in Figure 1d, the lower charge-transfer energy indicates an increased Ir–O covalency of Mo-PIO compared with PIO, manifesting increased activity of lattice oxygen (Table S3).

2). Page 5, line 114-117, one sentence is highlighted as follows.

It can be inferred that the decrease of O 2p band center is mainly caused by O₍₁₎ and O₍₂₎, and the enlarged Ir–O covalency, which facilitates the reactivity of lattice oxygen, mainly originates from the increase of Ir 5d band center (Figure 1d, Table S3).

3). Supporting Information, Table S3 is highlighted as follows.

Table S3. Band center and charge transfer energy determined by DOS

	$\epsilon_{\text{O-p}} / \text{eV}$	$\epsilon_{\text{Ir-d}} / \text{eV}$	$ \epsilon_{\text{Ir-d}} - \epsilon_{\text{O-p}} / \text{eV}$
Pr ₃ IrO ₇	-2.13	-2.45	0.32
Mo-Pr ₃ IrO ₇	-2.21	-2.09	0.12

Comment 7:

The schematic in Fig. 5e is impossible to understand. Where is the surface and what does the reconstruction look like? The same is true for many of the figures, like figure 7a or S12, where the viewing direction is unclear.

Response 7:

We thank the reviewer for the comment. We have checked and given the viewing direction of Figure 5e. The schematics in Figure 5e represent the evolution of the near surface region. The bulk is not shown after reconstructions. The annotations above each schematic briefly describe the reconstruction. The viewing direction in Figure 7a and Figure S16 has been given in the revised manuscript.

Modification 7:

1). **Figure 5e** is revised as follows.

Figure 5. Surface chemical states evolution in reconstruction. (a) XPS spectra of O 1s core-level with peak fitting results for PIO and 0.2Mo-PIO at varied electrochemical cycles. (b), (c) XPS spectra of Ir 4f core-level with peak fitting results for PIO (b) and 0.2Mo-PIO (c) at varied electrochemical cycles. (d) XPS spectra of Mo 3d core-level with peak fitting results for 0.2Mo-PIO at varied electrochemical cycles. (e) Schematic illustration of electrochemical surface reconstruction over Mo-Pr₃IrO₇.

2). **Figure 7a** is revised as follows.

Figure 7. Bridging-oxygen-assisted deprotonation pathway in OER. (a) Atomic structures of the (110) surfaces of IrO₂-O_v (left) and Mo-IrO₂-O_v. (b) The H atom adsorption energy on surface O_{br} sites in IrO₂-O_v and Mo-IrO₂-O_v. Right panel is the schematic illustration. (c) Schematic illustrations of the catalyst structure after reconstruction (left) and a faster BOAD process upon Mo substitution (right). (d) The free energy diagram of Mo-IrO₂-O_v with different OER pathways. PDS is labeled by green star. Color code: Ir (green), Mo (purple), and O (red). The green and purple octahedra represent [IrO₆] and [MoO₆] octahedrons, respectively.

3). Supporting Information, Figure S16 is revised as follows.

Figure S16. Crystal structure of IrO₂-O_v (a) and Mo doped IrO₂-O_v (b).

Comment 8:

In Fig. 8 the PDS should be highlighted to ease the discussion.

Response 8:

Thanks for the nice reminder from the reviewer. We have labeled the PDS in Figure 7d in the revised manuscript.

Modification 8:

1). Page 16, line 420-424, two sentences are revised as follows.

Furthermore, PDS locates at deprotonation of $\text{OO}_{\text{top}}^*-\text{OH}_{\text{bri}}^*$ with a large overpotential of 0.94 V (Figure S17, PDS is labeled by green star). As for Mo-doped IrO_2-O_v , with the introduction of BOAD pathway, the PDS of Mo-doped IrO_2-O_v shifts from $\text{OOH}_{\text{top}}^* \rightarrow \text{O}_2 + (\text{H}^+ + \text{e}^-)$ to $\text{OO}_{\text{top}}^* + \text{OH}_{\text{bri}}^* \rightarrow \text{O}_2 + (\text{H}^+ + \text{e}^-)$ with a decreased overpotential from 0.60 V to 0.43 V (Figure 7d, PDS is labeled by green star).

2). Figure 7d is revised as follows.

Figure 7. Bridging-oxygen-assisted deprotonation pathway in OER. (a) Atomic structures of the (110) surfaces of IrO_2-O_v (left) and $\text{Mo-IrO}_2-\text{O}_v$. (b) The H atom adsorption energy on surface O_{bri} sites in IrO_2-O_v and $\text{Mo-IrO}_2-\text{O}_v$. Right panel is the schematic illustration. (c) Schematic illustrations of the catalyst structure after reconstruction (left) and a faster BOAD process upon Mo substitution (right). (d) The free energy diagram of $\text{Mo-IrO}_2-\text{O}_v$ with different OER pathways. PDS is labeled by green star. Color code: Ir (green), Mo (purple), and O (red). The green and purple octahedra represent $[\text{IrO}_6]$ and $[\text{MoO}_6]$ octahedrons, respectively.

3). Supporting Information, Figure S17 is revised as follows.

Figure S17. The free energy diagram of IrO₂-O_v with different OER pathways. Color code: Ir (green), O (red). The green octahedra represent IrO₆ octahedra. PDS is labeled by the green star.

Comment 9:

When discussing the different OER pathways, the authors assume that chemical steps associated with the BOAD pathway are readily overcome. Is this really justified for steps of more than 1 eV? What temperatures would be required to enable these processes on a reasonable time scale?

Response 9:

We thank the reviewer for the comment. For the corrected Mo-doped IrO₂-O_v with PBE + U calculations, the most insuperable chemical steps have decreased to 0.46 eV. To verify the feasibility of the chemical steps, the rate constant was further discussed by the transition state theory (TST). In the TST, it is generally acknowledged that the TS locates at a point of no return between reactant state and product. And the transition of reactant to TS is the rate determining step (RDS). (Fundamental Concepts in Heterogeneous Catalysis; John Wiley & Sons, Inc: Hoboken, NJ, USA, 2014.) Based on this, the rate constant can be deduced as:

$$k_{\text{TST}} = \frac{k_{\text{B}}T}{h} e^{-\Delta G_{\text{TS}}/k_{\text{B}}T}$$

in which ΔG_{TS} is the difference of Gibbs energy between TS and the reactant state. This equation is the formula used in TST to calculate the rate constant of a reaction using thermodynamic methods. It can be applied to elementary reaction to calculate the reaction's rate constant so long as the activation

free energy (ΔG_{TS}) can be obtained. And the rate constant is equal to the reaction rate. As a rule of thumb, a reaction rate on the order of $1 \text{ site}^{-1} \text{ s}^{-1}$ should be attained for a reasonable catalyst. On the base of $1 \text{ site}^{-1} \text{ s}^{-1}$, Nørskov et al. depicted the relation of the ΔG_{TS} and the temperature (T) as shown in Figure S21. It reveals that the reaction rate of $1 \text{ site}^{-1} \text{ s}^{-1}$ corresponds to an ΔG_{TS} of 0.75 eV at room temperature (300 K). As a consequence, the activation free energy of 0.75 eV can be considered as a limit barrier for the chemical reaction steps at room temperature. Then we have calculated the free energy barrier (ΔG) of the chemical step: $\text{O}_{\text{top}}^* + \text{H}_2\text{O} \rightarrow \text{OOH}_{\text{top}}^* + \text{OH}_{\text{bri}}^*$. The corresponding ΔG_{TS} is 0.64 eV, which below 0.75 eV, indicating the kinetic barrier of chemical steps associated with the BOAD pathway can be readily overcome at room temperature.

Figure S21. The Gibbs free energy of activation, ΔG , plotted as a function of temperature for $r = 0.01 \text{ s}^{-1}$ (black curve), $r = 1 \text{ s}^{-1}$ (light gray curve), and $r = 100 \text{ s}^{-1}$ (dark gray curve) as calculated using the equation above.²¹ Copyright 2014, John Wiley & Sons.

21. Nørskov JK, Studt F, Abild-Pedersen F, Bligaard T. *Fundamental concepts in heterogeneous catalysis*. John Wiley & Sons (2014).

Modification 9:

1). Page 16, line 426-431, two sentences are added as follows.

To verify the feasibility of the chemical steps, the rate constant of the second H_2O dissociation ($\text{O}_{\text{top}}^* + \text{H}_2\text{O} \rightarrow \text{OOH}_{\text{top}}^* + \text{OH}_{\text{bri}}^*$) was further discussed by the transition state theory (TST). According to the climbing image nudged elastic band (CINEB) calculation,⁶⁷ its corresponding activation free energy barrier is demonstrated as 0.64 eV (Figure S18), which is below the hard limit for a surmountable barrier (0.75 eV) at room temperature, indicating that the chemical steps in BOAD can process readily (Figure S21, Supporting Note 6).⁶⁸

67. Henkelman G, Uberuaga BP, Jónsson H. A climbing image nudged elastic band method for finding saddle points and minimum energy paths. *J. Chem. Phys.* **113**, 9901-9904 (2000).

68. Nørskov JK, Studt F, Abild-Pedersen F, Bligaard T. *Fundamental concepts in heterogeneous catalysis*. John Wiley & Sons (2014).

2). **Supporting Information**, one sentence is added in **Computational Methods**.

Climbing image nudged elastic band (CINEB) was used for calculating the free energy barrier (ΔG_{TS}) of transition state⁹

9. Henkelman G, Uberuaga BP, Jónsson H. A climbing image nudged elastic band method for finding saddle points and minimum energy paths. *J. Chem. Phys.* **113**, 9901-9904 (2000).

2). **Supporting Information**, **Figure S18** is added as follows.

Figure S18. Activation free energy (ΔG_{TS}) for the second H₂O dissociation in BOAD pathway. TS is the transition state, the ΔG_{TS} is 0.64 eV.

3). **Supporting Information**, **Supporting Note 6** is added as follows.

Supporting Note 6

In the transition state theory (TST), it is generally acknowledged that the TS locates at a point of no return between reactant state and product. And the transition of reactant to TS is the rate determining step (RDS).²¹ Based on this, the rate constant can be deduced as:

$$k_{TST} = \frac{k_B T}{h} e^{-\Delta G_{TS}/k_B T}$$

in which ΔG_{TS} is the difference of Gibbs energy between TS and the reactant state. This equation is the formula used in TST to calculate the rate constant of a reaction using thermodynamic methods. It can be applied to elementary reaction to calculate the reaction's rate constant so long as the activation free energy (ΔG_{TS}) can be obtained. And the rate constant is equal to the reaction rate. As a rule of thumb, a reaction rate on the order of 1 site⁻¹ s⁻¹ should be attained for a reasonable catalyst. On the base of 1 site⁻¹ s⁻¹, Nørskov et al. depicted the relation of the ΔG_{TS} and the temperature (T) as shown in Figure S21.²¹ It reveals that the reaction rate of 1 site⁻¹ s⁻¹ corresponds to an ΔG_{TS} of 0.75 eV at room temperature (300 K). As a consequence, the activation free energy of 0.75 eV can be considered

as a limit barrier for the chemical reaction steps at room temperature. Then we calculated the free energy barrier (ΔG) of the chemical step: $O_{top}^* + H_2O \rightarrow OOH_{top}^* + OH_{bri}^*$. The corresponding ΔG_{TS} is 0.64 eV, which below 0.75 eV, indicating the kinetic barrier of chemical steps associated with the BOAD pathway can be readily overcome at room temperature.

Figure S21. The Gibbs free energy of activation, ΔG , plotted as a function of temperature for $r = 0.01 \text{ s}^{-1}$ (black curve), $r = 1 \text{ s}^{-1}$ (light gray curve), and $r = 100 \text{ s}^{-1}$ (dark gray curve) as calculated using the equation above.²¹ Copyright 2014, John Wiley & Sons.

21. Nørskov JK, Studt F, Abild-Pedersen F, Bligaard T. *Fundamental concepts in heterogeneous catalysis*. John Wiley & Sons (2014).

Reviewer #2

In this work, the authors have demonstrated $Pr_3Ir_{1-x}Mo_xO_7$ with high-valence Mo modulated to be a good catalyst toward acidic OER. The accelerated surface reconstruction with Mo doping also leads to buffered charge compensation, which prevents the fierce Ir dissolution and excessive lattice oxygen loss. The results are interesting and supported by comprehensive theoretical predictions and solid experimental evidences. However, there are still some issues to be addressed before being publishing in Nature Communications.

Please find the comments below.

Comment 1:

The KIE analysis in Fig.6d is not rigorous at the current state. First, only the H_2O was replaced by the D_2O , however, the $HClO_4$ was used in both electrolyte; the H is not completely substituted by D. Secondly, the potential scale was normalized on the RHE scale, rather the overpotential scale. Actually, the pD/pH, and the equilibrium potential for D_2O/O_2 and H_2O/O_2 are all different. Please check the literature and do a more rational analysis.

Response 1:

Thanks for the rigorous and valuable comment from the reviewer. Actually, we tried to buy deuterated perchloric acid (DClO₄) at first but did not make it. Since DClO₄ is not commonly used, many chemical reagent suppliers claimed to be out of stock or would not offer its production. We calculated the mole fraction of ¹H in all hydrogen to be only 0.0913% (Supporting Note 4), which has negligible effect on the experiment. So HClO₄ was used for KIE analysis in the submitted manuscript.

We have been aware of the different equilibrium potential for D₂O/O₂ (1.262 V) and H₂O/O₂ (1.229 V) and converted the potential to overpotential based on different equilibrium potential correctly.

Although DClO₄ was not available, we got deuterated sulfuric acid (D₂SO₄). To eliminate the influence of ¹H completely, KIE analysis was also conducted in 0.5M H₂SO₄ (in H₂O) and 0.5M D₂SO₄ (in D₂O) (Figure S14). The conclusion drawn from the two sets of experiments are consistent that proton activity has a greater impact on the OER of PIO-post than 0.2Mo-PIO-post.

Modification 1:

1). Page13, line 341-344, two sentences are revised and three references are added as follows.

OER activities were also tested in proton (0.1 M HClO₄ in H₂O) and deuterium (0.1 M HClO₄ in D₂O) electrolytes to investigate the kinetic isotope effect (KIE) (Figure 6c, Supporting Note 4).^{59, 60, 61} KIE values were obtained based on the ratio of OER current density for hydrogen to that for deuterium at the same overpotential (Figure 6d).

59. Malko D, Kucernak A. Kinetic isotope effect in the oxygen reduction reaction (ORR) over Fe-N/C catalysts under acidic and alkaline conditions. *Electrochem. Commun.* **83**, 67-71 (2017).

60. Huang J, *et al.* Modifying redox properties and local bonding of Co₃O₄ by CeO₂ enhances oxygen evolution catalysis in acid. *Nat. Commun.* **12**, 3036-3046 (2021).

61. He Z, *et al.* Activating lattice oxygen in NiFe-based (oxy)hydroxide for water electrolysis. *Nat. Commun.* **13**, 2191-2202 (2022).

2). Figure 6c-d are revised as follows.

Figure 6. Role of reconstructed Ir–O_{bri}–Mo species in promoting deprotonation. (a) LSV curves for PIO-post and 0.2Mo-PIO-post in HClO₄ with varied pH. (b) OER current density at 1.53 V_{RHE} plotted in log scale as a function of pH, from which the proton reaction orders ($\rho^H = \partial \log(j) / \partial \text{pH}$) were calculated. (c) LSV curves for PIO-post and 0.2Mo-PIO-post measured in 0.1 M HClO₄ prepared in H₂O and D₂O. (d) KIE of PIO-post and 0.2Mo-PIO-post. j^H and j^D are referred to the current density measured in 0.1M HClO₄ prepared in H₂O and D₂O at the same overpotential, respectively. (e) *In situ* Raman spectra of PIO-post (red) and 0.2Mo-PIO-post (blue) in 0.1M HClO₄ at varied potentials.

3). Supporting Information, Supporting Note 4 is added as follows.

Supporting Note 4: Calculation of the mole fraction of ¹H in all hydrogen

We calculated the mole fraction of ¹H in all hydrogen to elucidate that ¹H in nondeuterated HClO₄ will not affect the results of the experiment significantly.

Take 1 L 0.1 M HClO₄ (in D₂O) as an example.

The total amount of ¹H is 0.1 mole. The volume of D₂O used is

$$1000 \text{ mL} - 8.625 \text{ mL} = 991.375 \text{ mL}$$

where 8.625 mL is the volume of HClO₄ added.

The total amount of D is

$$991.375 \text{ mL} \times 1.1056 \text{ g mL}^{-1} \div 20.0276 \text{ g mol}^{-1} \times 2 = 109.4554 \text{ mol}$$

where 1.1056 g mL⁻¹ is the density of D₂O, 20.0276 g mol⁻¹ is the molecular weight of D₂O.

So the mole fraction of ¹H is

$$\frac{0.1 \text{ mol}}{0.1 \text{ mol} + 109.4554 \text{ mol}} \times 100\% = 0.0913\%$$

The mole fraction of ^1H is so small that it has negligible effect on the experiment. So HClO_4 was used for KIE analysis.

4). Page13, line 347-349, one sentence is added as follows.

The deuterium isotopic experiments conducted in 0.5 M H_2SO_4 (in H_2O) and 0.5M D_2SO_4 (in D_2O) also provided consistent results (Figure S14).

5). Supporting Information, Figure S14 is added as follows.

Figure S14. (a) LSV curves for PIO-post and 0.2Mo-PIO-post measured in 0.5 M H_2SO_4 prepared in H_2O and 0.5 M D_2SO_4 prepared D_2O . (d) KIE of PIO-post and 0.2Mo-PIO-post. j^{H} and j^{D} are referred to the current density measured in 0.1M HClO_4 prepared in H_2O and D_2O , respectively.

Comment 2:

The authors selected weberite type Pr_3IrO_7 material systems for acid OER research to study the effect of Mo doping on catalyst reconstruction and active substances. As revealed by the authors, the active substance is Ir-O-Mo motif. Then this brings up a curiosity about whether Mo doping also has a positive effect in the IrO_2 system. The universality of this Mo doping should be verified in other systems.

Response 2:

Thanks for the insightful comment from the reviewer. We have synthesized IrO_2 and Mo doped IrO_2 (Mo- IrO_2) by sol-gel method. Structural characterizations and electrochemical tests show that Mo has been successfully introduced into IrO_2 system. And importantly, Mo- IrO_2 exhibits better OER activity than IrO_2 in 0.1 M HClO_4 , manifesting that Mo doping indeed has a positive effect in IrO_2 system (Figure S19, Supporting Note 7). As for other systems, we chose another representative electrocatalyst RuO_2 to investigate whether Mo doping (Mo- RuO_2) plays a positive role. As shown in the revised

manuscript, Mo has significant influence on the OER behavior of RuO₂ in 0.1 M HClO₄. Mo doping not only contributes to higher OER activity but importantly, the excessive oxidation at higher potentials of Mo-RuO₂ has been obviously improved, which is more important for practical applications (Figure S20, Supporting Note 8). Our investigations suggest that in addition to weberite type Pr₃IrO₇ system, Mo also plays a positive role in IrO₂ and RuO₂ systems in acidic OER.

Modification 2:

1). Page 16, line 437-441, one paragraph is added as follows.

To verify the universality of Mo doping in regulating OER performance, we synthesized Mo doped IrO₂ (Mo-IrO₂) and Mo doped RuO₂ (Mo-RuO₂) via sol-gel method (see Supporting Information for details) and further compare the electrochemical performance with their undoped counterparts. It is found that Mo also improves the performance of IrO₂ and RuO₂ systems in acidic OER (Figure S19-S20, Supporting Note 7-8).

2). Supporting Information, Page 2, one paragraph is added as follows.

As comparison, Mo-doped IrO₂ precursor was synthesized by sol-gel method and then calcined in air to obtain the oxide powder. For the synthesis of precursor, 0.025 g molybdenum(V) chloride (Macklin, 99.5%) was firstly dissolved in 4 mL ethylene glycol (J&K Scientific, 99%, extra pure), then 0.12 g iridium (IV) chloride hydrate (Macklin, Ir 48.0 - 55.0%) was added. After dissolution, 0.16 g citric acid (J&K Scientific, 99.5%, anhydrous, ACS reagent) which has been beforehand dissolved in 15 mL water and 0.1 mL ammonia solution (Aladdin, GR, 25-28%) was dropped into above solution. The resulting solution was stirred at 90 °C for 8 h, and then heated in an oven at 170 °C for 12 h to obtain a solid precursor. Then, this precursor was calcined in air at 550 °C for 6 h to acquire the targeted catalysts. Undoped IrO₂ was synthesized without adding molybdenum(V) chloride. Mo-doped RuO₂ was synthesized by the same method as Mo-doped IrO₂ except for choosing ruthenium (III) chloride (J&K Scientific, 99%, anhydrous) rather than iridium (IV) chloride hydrate. Undoped RuO₂ was synthesized without adding molybdenum(V) chloride following the same procedure as Mo-RuO₂.

3). Supporting Information, Figure S19 is added as follows.

Figure S19. (a) Powder XRD patterns of IrO₂ and Mo-IrO₂. (b), (c) HRTEM images of IrO₂ (b) and Mo-IrO₂ (c). Insets: corresponding EDS elemental mappings of Ir (purple), Mo (orange) and O (blue). (d) - (f) OER performance tests of IrO₂ and Mo-IrO₂ conducted in 0.1 M HClO₄. (d) Geometric area-normalized LSV curves. (e) Tafel plots based on geometric area-normalized LSV curves. (f) Nyquist plots at 1.55 V_{RHE}.

4). Supporting Information, Figure S20 is added as follows.

Figure S20. (a) Powder XRD patterns of RuO₂ and Mo-RuO₂. (b), (c) HRTEM images of RuO₂ (b) and Mo-RuO₂ (c). Insets: corresponding EDS elemental mappings of Ru (green), Mo (orange) and O (blue). (d) - (f) OER performance tests of RuO₂ and Mo-RuO₂ conducted in 0.1 M HClO₄. (d) Geometric area-normalized LSV curves at specific cycles. (e) Tafel plots based on geometric area-normalized LSV curves. (f) Nyquist plots at 1.53 V_{RHE}.

5). **Supporting Information, Supporting Note 7** is added as follows.

Supporting Note 7

As shown in Figure S19a, powder XRD patterns of as synthesized IrO₂ and Mo-IrO₂ can be well indexed to rutile IrO₂ (PDF # 97-005-6009). HRTEM of IrO₂ (Figure S19b) and Mo-IrO₂ (Figure S19c) confirm the good crystallization. The interplanar spacing of about 0.259 nm in Figure S19b and 0.319 nm in Figure S19c coincide well with (101) and (110) facets of rutile IrO₂ respectively. Besides, no impurity particles are observed in Mo-IrO₂ and EDS elemental mapping results (insets) clearly show the homogeneous dispersion of Mo in the lattice. These results demonstrate that Mo atoms are successfully introduced into IrO₂ lattice with well-maintained crystallization. OER performance was assessed in 0.1 M HClO₄ electrolyte under ambient conditions. Mo-IrO₂ exhibits an overpotential of 320 mV to reach 10 mA cm_{geo}⁻², lower than that of IrO₂ (350 mV) (Figure S19d). Tafel slope of Mo-IrO₂ (62.41 mV dec⁻¹) is smaller than undoped IrO₂ (74.44 mV dec⁻¹), suggesting the optimized kinetics of IrO₂ after Mo substitution (Figure S19e). The decreased semicircles in Nyquist plots (Figure S19f) clearly demonstrate a facilitated charge transfer process of Mo-IrO₂. These results manifest that Mo can be successfully introduced into IrO₂ system and indeed has a positive effect during acidic OER.

6). **Supporting Information, Supporting Note 8** is added as follows.

Supporting Note 8

As shown in Figure S20a, powder XRD patterns of as synthesized RuO₂ and Mo-RuO₂ can be well indexed to rutile RuO₂ (PDF # 97-005-6007). HRTEM of RuO₂ (Figure S20b) and Mo-RuO₂ (Figure S20c) confirm the good crystallization. The interplanar spacing of about 0.256 nm in Figure S20b and 0.321 nm in Figure S20c coincide well with (101) and (110) facets of rutile RuO₂ respectively. Besides, no impurity particles are observed in Mo-RuO₂ and EDS elemental mapping results (insets) clearly show the homogeneous dispersion of Mo in the lattice. These results demonstrate that Mo atoms are successfully introduced into RuO₂ lattice with well-maintained crystallization. OER performance assessment in 0.1 M HClO₄ demonstrates that Mo has significant influence on the behavior of RuO₂. The activity of RuO₂ deteriorates quickly after one LSV because of the excessive oxidation of Ru species at high potentials. While the polarization curves of Mo-RuO₂ exhibit no decay after 20 cycles (Figure S20d), suggesting the improved stability upon Mo substitution. Mo-RuO₂ requires an overpotential of 290 mV to reach 20 mA cm_{geo}⁻², lower than that of RuO₂ (360 mV). Tafel slope of

Mo-RuO₂ (46.89 mV dec⁻¹) is smaller than undoped RuO₂ (67.86 mV dec⁻¹), suggesting the optimized kinetics (Figure S20e). The decreased semicircles in Nyquist plots (Figure S20f) clearly demonstrate a facilitated charge transfer process of Mo-RuO₂. Consequently, Mo doping not only contributes to higher OER activity but importantly, the excessive oxidation at higher potentials of Mo-RuO₂ has been obviously impeded, which is more important for practical applications.

Comment 3:

In this work, OER performance was assessed in 0.1M HClO₄ electrolyte, which is different from those in literature (usually pH ≈ 0). It would be helpful to provide the OER performance tested in 1 M HClO₄ or 0.5 M H₂SO₄, and then compared to other acidic OER catalysts.

Response 3:

We thank the reviewer for this comment. We have conducted OER performance tests in 0.5 M H₂SO₄ and the comparisons with other reported Ir-based electrocatalysts are provided based on geometric and mass activities.

Modification 3:

1). Page 8, line 208-210, one sentence is added as follows.

And OER performance tests conducted in 0.5 M H₂SO₄ further reveal that 0.2Mo-PIO-post is among the most active electrocatalysts for OER in acid (Figure S9, Table S12).

2). Supporting Information, Figure S9 is added as follows.

Figure S9. OER performance tests of x Mo-PIO-post ($x = 0.0, 0.2$) and commercial IrO₂ conducted in 0.5 M H₂SO₄. (a) Geometric area-normalized LSV curves. (b) Tafel plots based on geometric area-normalized LSV curves. (c) Mass loading-normalized LSV curves.

3). Supporting Information, Table S12 is added as follows.

Table S12. The comparison of OER performance with some representative acidic OER electrocatalysts in 0.5 M H₂SO₄.

Catalysts	η_{10} (mV)	Mass loading	Mass activity	Tafel slope (mV·dec ⁻¹)	Ref.
0.2Mo-PIO-post	265	0.25 mg·cm ⁻²	1216 A·g _{Ir} ⁻¹ @1.55V	62.15	This work
PIO-post	303	0.25 mg·cm ⁻²	308A·g _{Ir} ⁻¹ @1.55V	76.68	This work
IrO ₂	337	0.25 mg·cm ⁻²	33 A·g _{Ir} ⁻¹ @1.55V	70.21	This work
W-Ir-B alloy	300	78.9 μg·cm ⁻²	518 A·g _{Ir} ⁻¹ @1.53V	78	Nat. Commun. 12 , 3540 (2021).
AlNiCoIrMo np- HEA (20 wt%)	275		115 A·g ⁻¹ @1.5 V	55.2	Small 15 , 1904180 (2019).
Li-IrO _x	290	0.125 mg·cm ⁻²	100 A·g _{Ir} ⁻¹ @1.52V	39	J. Am. Chem. Soc. 141 , 3014 (2019).
Ru-N-C	267	0.28 mg·cm ⁻²	3571 A·g _{Ru} ⁻¹ @1.497 V	52.6	Nat. Commun. 10 , 4849 (2019).
Co ₃ O ₄ /CeO ₂	423			88.1	Nat Commun 12 , 3036 (2021).
Ir/Fe ₄ N	316 ± 5	76.5 μg·cm ⁻²	116.4 mA·μg _{Ir} ⁻¹ @1.54 V	61.5	ACS Catal. 8 , 2615 (2018).
Rh ₂₂ Ir ₇₈ NPs	292 ± 1	0.28 mg·cm ⁻²	1174 ±20 A·g _{Ir} ⁻¹ @1.53 V	101	ACS Nano 13 , 13225 (2019).
IrO _x /SrIrO ₃	270-290				Science 353 , 1011 (2016).
Ir@N-G-750	303	23 μg·cm ⁻²	2420 A·g ⁻¹ @1.6 V	50	Nano Energy 62 , 117 (2019).
IrO ₂ @Ir/TiN	265	0.379 mg·cm ⁻²	480.4 A·g _{Ir} ⁻¹ @1.6 V	52.3	ACS Appl. Mater. Interfaces 10 , 38117 (2018).
6H-SrIrO ₃	248	0.9 mg·cm ⁻²	75 A·g _{Ir} ⁻¹ @1.525 V		Nat. Commun. 9 , 5236 (2018).
TiN/IrO ₂	313		874 A g _{IrO₂} ⁻¹ @1.6 V	65.5	J. Mater. Sci. 55 , 3507 (2020).

Comment 4:

The stability measurement is conducted at the geometric current density of 5 mA cm⁻², which is quite low compared to the results in the literature. The measurement on higher current density should be provided to make a fair comparison. Besides, the Ir leaching should be also investigated. The stability number should be calculated and compared to other Ir-based catalysts in the literature.

Response 4:

We thank the reviewer for the comment. To make a fair comparison, we have conducted the stability test for 0.2Mo-PIO and commercial IrO₂ at the geometric current density of 20 mA cm⁻² and further evaluated the stability number (S-number) based on ICP-MS results. Besides, comparison of S-number with other Ir-based catalysts has been listed in the revised manuscript.

Modification 4:

1). Page 9, line 242-248, two sentences are revised as follows.

Besides, 0.2Mo-PIO exhibits excellent stability with an overpotential increment less than 50 mV at 5 mA cm⁻² after 200 hours, while PIO deteriorates quickly within 20 hours (Figure 3f). Chronopotentiometry curves at 20 mA cm⁻² manifests the better stability of 0.2Mo-PIO (overpotential increment less than 30 mV after 140 hours) than commercial IrO₂ (Figure S12), as further verified by the higher stability number (S-number) of 2.1×10^8 (see Supporting Information for details, comparisons of stability number with other Ir-based electrocatalysts are listed in Table S16).

2). Supporting Information, Figure S12 is added as follows.

Figure S12. Chronopotentiometry curves of 0.2Mo-PIO and commercial IrO₂ at 20 mA cm⁻² in 0.1 M HClO₄ electrolyte. The abrupt changes of voltage are caused by refreshing the electrolyte and the jagged voltage fluctuation is due to the release of bubbles accumulating on the electrode surface.

3). Supporting Information, Table S16 is added as follows.

Table S16. The comparison of stability number with some representative Ir-based electrocatalysts in acidic media.

Catalysts	Electrochemical testing conditions	Electrolyte	S-number	Ref.
0.2Mo-PIO-post	20 mA cm ⁻² , 140 h	0.1 M HClO ₄	2.1×10^8	This work

Catalysts	Electrochemical testing conditions	Electrolyte	S-number	Ref.
commercial IrO ₂	20 mA cm ⁻² , 20 h	0.1 M HClO ₄	6.5 × 10 ⁷	This work
IrO _x	5 mV s ⁻¹ sweep to 1.65 V _{RHE}	0.1 M HClO ₄	5.7 × 10 ⁴	Nat. Catal. 1 , 508 (2018).
SrIrO ₃	5 mV s ⁻¹ sweep to 1.65 V _{RHE}	0.1 M HClO ₄	1.8 × 10 ⁴	Nat. Catal. 1 , 508 (2018).
SrIr _{0.8} Zn _{0.2} O ₃	1.7 V _{RHE} , 160 min	0.1 M HClO ₄	3.2 × 10 ⁵	ACS Appl. Energy Mater 5 , 12206 (2022).
IrO _x	100 A/g _{Ir} , 2 h	0.1 M H ₂ SO ₄	6.0 × 10 ⁴	Nat. Commun. 12 , 1 (2021).
porous IrO _x -500 °C	10 mA cm ⁻² , V _{cutoff} = 2 V _{RHE}	0.05 M H ₂ SO ₄	3.5 × 10 ⁴	ACS Catal. 11 , 4107 (2021).
SrCo _{0.9} Ir _{0.1} O _{3-δ}	10 mA cm ⁻² , 3 h	0.1 M HClO ₄	8.6 × 10 ⁴	Nat. Commun. 10 , 1 (2019).
SrIrO ₃ STO	1.7 V _{RHE} average, 6 h	0.5 M H ₂ SO ₄	9.8 × 10 ³	Adv. Funct. Mater. 31 , 2101542 (2021).
BCC-Cr-SrIrO ₃	1.53 V _{RHE} , 40 h	0.1 M HClO ₄	6.5 × 10 ⁵	Nano Energy 102 , 107680 (2022).
3C-SrIrO ₃	1.53 V _{RHE} , 40 h	0.1 M HClO ₄	2.5 × 10 ⁴	Nano Energy 102 , 107680 (2022).
6H-SrIrO ₃	1 mA cm ⁻² , 1 h	0.1 M HClO ₄	2.2 × 10 ⁴	Chem. Mater. 32 , 3499 (2020).
Y ₂ Ir ₂ O ₇	1.6 V _{RHE} , ~2.5 h	0.5 M H ₂ SO ₄	5.5 × 10 ³	J. Phys. Chem. C 126 , 1751 (2022).
Ir/CuTiON _x /C	5 mA cm ⁻² , 5 min	0.1 M HClO ₄	7.8 × 10 ⁴	ACS Catal. 11 , 12510 (2021).
Re _{0.1} -IrO ₂	10 mA cm ⁻² , 30 h	0.5 M H ₂ SO ₄	3.7 × 10 ¹⁰	Small 2207847 (2023).

4). **Supporting Information, Page 3**, one paragraph and one reference are added as follows.

1.4. Stability Number (S-number) Calculation. The S-number was calculated by the following equation as previously reported:¹

$$\text{S-number} = \frac{n_{\text{O}_2}}{n_{\text{Ir (dissolved)}}$$

where n_{O2} and n_{Ir (dissolved)} refer to the total amount of evolved oxygen (calculated from total charge)

during the chronopotentiometry test and the amount of dissolved Ir extracted from ICP-MS results, respectively.

1. Geiger S, *et al.* The stability number as a metric for electrocatalyst stability benchmarking. *Nat. Catal.* **1**, 508-515 (2018).

Comment 5:

In introduction, authors claimed that “the mass content of iridium in Ln_3IrO_7 compounds (26.4% in Pr_3IrO_7) is obviously lower than that in IrO_2 (85.7%), perovskite and pyrochlore structures (58.6% in SrIrO_3 and 49.4% in $\text{Pr}_2\text{Ir}_2\text{O}_7$). Consequently, if properly tuned, directional surface reconstruction with enhanced stability and further decreased iridium consumption of Ln_3IrO_7 will be anticipated.” It is not rigorous, since in the PEM electrolytic water system, we are more concerned about the loading amount of precious metals on the membrane electrode. It is more convincing to compare the mass activity (normalized to Ir mass) of the as-prepared catalysts herein to other Ir-based catalysts.

Response 5:

Thanks for the valuable comment from the reviewer. We have revised this sentence and compared the mass activity of catalysts herein with other Ir-based electrocatalysts in acidic media.

Modification 5:

- 1). Page 2, line 52-53, one sentence is revised as follows.

Consequently, if properly tuned, directional surface reconstruction with improved stability and mass activity (normalized to Ir mass) of Ln_3IrO_7 will be anticipated.

- 2). Page 7, line 204-208, two sentences are added as follows.

As shown in Figure S8, 0.2Mo-PIO-post gives a mass activity of $415 \text{ A} \cdot \text{g}_{\text{Ir}}^{-1}$ at 1.52 V, about 4.3 and 88.3 times higher than that of PIO-post ($97 \text{ A} \cdot \text{g}_{\text{Ir}}^{-1}$) and commercial IrO_2 ($4.7 \text{ A} \cdot \text{g}_{\text{Ir}}^{-1}$). The iridium mass activity was also compared with reported Ir-based catalysts in acidic media (Table S11), demonstrating the high mass activity of 0.2Mo-PIO-post among reported Ir-based catalysts.

- 3). Supporting Information, Figure S8 is added as follows.

Figure S8. Mass loading-normalized LSV curves of $x\text{Mo-PIO-post}$ ($x = 0.0, 0.2$) and commercial IrO_2 in 0.1 M HClO_4 .

4). **Supporting Information, Table S11** is added as follows.

Table S11. The comparison of OER performance with some representative Ir-based electrocatalysts in acidic media.

Catalysts	Electrolytes	Mass loading	Mass activity	Ref.
0.2Mo-PIO-post	0.1 M HClO_4	$0.25 \text{ mg} \cdot \text{cm}^{-2}$	$415 \text{ A} \cdot \text{g}_{\text{Ir}}^{-1} @ 1.52 \text{ V}$	This work
PIO-post	0.1 M HClO_4	$0.25 \text{ mg} \cdot \text{cm}^{-2}$	$97 \text{ A} \cdot \text{g}_{\text{Ir}}^{-1} @ 1.52 \text{ V}$	This work
IrO_2	0.1 M HClO_4	$0.25 \text{ mg} \cdot \text{cm}^{-2}$	$4.7 \text{ A} \cdot \text{g}_{\text{Ir}}^{-1} @ 1.52 \text{ V}$	This work
$\text{Pt}_{39}\text{Ir}_{10}\text{Pd}_{11}$	0.1 M HClO_4	$16.8 \mu\text{g Pt+Ir+Pd} \cdot \text{cm}^{-2}$	$200 \text{ A} \cdot \text{g}_{\text{Pt+Ir+Pd}}^{-1} @ 1.53 \text{ V}$	Adv. Energy Mater. 10 , 1904114 (2020).
9R-Ba IrO_3	0.5 M H_2SO_4	$0.28 \text{ mg} \cdot \text{cm}^{-2}$	$168 \text{ A} \cdot \text{g}_{\text{Ir}}^{-1} @ 1.5 \text{ V}$	J. Am. Chem. Soc. 143 , 43, 18001 (2021).
P-Ir $\text{Cu}_{1.4}$ NCs	0.05 M H_2SO_4	$60 \mu\text{g Ir} \cdot \text{cm}^{-2}$	$213 \text{ A} \cdot \text{g}_{\text{Ir}}^{-1} @ 1.55 \text{ V}$	Chem. Mater. 30 , 8571 (2018).
6H-Sr IrO_3	0.5 M H_2SO_4	$0.9 \text{ mg} \cdot \text{cm}^{-2}$	$75 \text{ A} \cdot \text{g}_{\text{Ir}}^{-1} @ 1.525 \text{ V}$	Nat. Commun. 9 , 5236 (2018).
IrNiCu DNF/C	0.1 M HClO_4	$20 \mu\text{g} \cdot \text{cm}^{-2}$	$460 \pm 70 \text{ A} \cdot \text{g}_{\text{Ir}}^{-1} @ 1.53 \text{ V}$	ACS Nano 11 , 5500 (2017).
cobalt-doped 6H-Sr IrO_3	0.1 M HClO_4	$0.45 \text{ mg} \cdot \text{cm}^{-2}$	$286.7 \text{ A} \cdot \text{g}_{\text{Ir}}^{-1} @ 1.55 \text{ V}$	ACS Appl. Mater. Interfaces 11 , 42006 (2019).
$\text{Y}_2[\text{Ru}_{1.6}\text{Y}_{0.4}]\text{O}_{7-\delta}$	0.1 M HClO_4		$150 \text{ A} \cdot \text{g}^{-1} @ 1.45 \text{ V}$	Angew. Chem. Int. Ed. 130 , 14073 (2018).

Catalysts	Electrolytes	Mass loading	Mass activity	Ref.
Pr ₂ Ir ₂ O ₇	0.1 M HClO ₄	0.057 mg·cm ⁻²	424.5 A·g _{Ir} ⁻¹ @1.53 V	Adv. Mater. 31 , 1805104 (2019).
Amorphous Ir nanosheets	0.1 M HClO ₄	0.2 mg·cm ⁻²	221.8 A·g ⁻¹ @1.53 V	Nat. Commun. 10 , 4855 (2019).
Co-doped IrCu	0.1 M HClO ₄	25.5 μg·cm ⁻²	640 A·g _{Ir} ⁻¹ @1.53 V	Adv. Funct. Mater. 27 , 1604688 (2017).
Pd@Ir ₃ L	0.1 M HClO ₄	10.2 μg _{Ir} ·cm ⁻²	333 A g _{Ir} ⁻¹ @1.53 V	Chem. Mater. 31 , 5867 (2019).
Ba ₂ PrIrO ₆	0.1 M HClO ₄	0.95 mg·cm ⁻²	10 A g ⁻¹ @1.5 V	Nat. Commun. 7 , 12363 (2016).
IrO ₂ Nanoneedles	1 M H ₂ SO ₄	0.25 mg·cm ⁻²	50 A·g _{Ir} ⁻¹ @1.55 V	Adv. Funct. Mater. 28 , 1704796 (2018).

Comment 6:

The reference citation in the following sections is insufficient.

On line 179-187, the discussions on Figure 2e.

On line 213-218, “Two successive oxidation peaks...” and “...a deprotonation process of active oxygen intermediates regulated by Mo.”.

On line 228-229, “...probably modified OER pathway...”.

In the part of discussing about Mo-buffered charge compensation during reconstruction, there is a lack of extensive reference here.

Response 6:

We thank the reviewer for the comment. We have added references in corresponding sections in the revised manuscript.

Modification 6:

1a). Page 7, line 183-184, two references are added as follows.

O 1s spectra were fitted into four components (Figure 2e, Supporting Note 2).^{29, 35}

29. Diaz-Morales O, *et al.* Iridium-based double perovskites for efficient water oxidation in acid media. *Nat. Commun.* **7**, 12363-12368 (2016).

35. Xu X, *et al.* A perovskite electrocatalyst for efficient hydrogen evolution reaction. *Adv. Mater.* **28**, 6442-6448 (2016).

1b). Page 7, line 186-188, three references are added as follows.

Mo doping renders slightly positive-shifted binding energies for both of O_{L2} and $-OH/O_{ads}$ accompanying higher contribution from O_{L2} but lower from $-OH/O_{ads}$, indicating highly activated surface with a large amount of electrophilic oxygen species.^{36, 37, 38}

36. Pfeifer V, *et al.* In situ observation of reactive oxygen species forming on oxygen-evolving iridium surfaces. *Chem. Sci.* **8**, 2143-2149 (2017).

37. Pfeifer V, *et al.* Reactive oxygen species in iridium-based OER catalysts. *Chem. Sci.* **7**, 6791-6795 (2016).

38. Nong HN, *et al.* A unique oxygen ligand environment facilitates water oxidation in hole-doped $IrNiO_x$ core-shell electrocatalysts. *Nat. Catal.* **1**, 841-851 (2018).

1c). Page 7, line 190-191, two references are added as follows.

The stronger signal intensities at $g=1.998$ for Mo-doped samples than that of PIO agree well with the results from XPS analysis (Figure 2g).^{39, 40}

39. Kaftelen H, Ocakoglu K, Thomann R, Tu S, Weber S, Erdem E. EPR and photoluminescence spectroscopy studies on the defect structure of ZnO nanocrystals. *Phys. Rev. B* **86**, 014113 (2012).

40. Wang S, *et al.* Titanium-defected undoped anatase TiO_2 with p-type conductivity, room-temperature ferromagnetism, and remarkable photocatalytic performance. *J. Am. Chem. Soc.* **137**, 2975-2983 (2015).

2a). Page 8, line 223-225, two references are added as follows.

Two successive oxidation peaks at about 0.9 V and 1.2 V in PIO-post are corresponding to $Ir^{III/IV}$ and $Ir^{IV/V}$ species, suggesting the formation of Ir-enriched surfaces with highly active species.^{12, 41, 42}

12. Chen Y, *et al.* Exceptionally active iridium evolved from a pseudo-cubic perovskite for oxygen evolution in acid. *Nat. Commun.* **10**, 572-581 (2019).

41. Minguzzi A, *et al.* Easy Accommodation of Different Oxidation States in Iridium Oxide Nanoparticles with Different Hydration Degree as Water Oxidation Electrocatalysts. *ACS Catal.* **5**, 5104-5115 (2015).

42. Zhao F, *et al.* Increasing iridium oxide activity for the oxygen evolution reaction with hafnium modification. *J. Am. Chem. Soc.* **143**, 15616-15623 (2021).

2b). Page 8, line 225-228, three references are added as follows.

As for 0.2Mo-PIO-post, however, profile of CV curves is broader and the oxidation peak at 0.98 V generally considered as deprotonation of two-coordinated bridge oxygen is much more dominant, indicating a deprotonation process of active oxygen intermediates regulated by Mo.^{22, 43, 44}

22. Rao RR, *et al.* Towards identifying the active sites on $RuO_2(110)$ in catalyzing oxygen evolution. *Energy Environ. Sci.* **10**, 2626-2637 (2017).

43. Rao RR, *et al.* Operando identification of site-dependent water oxidation activity on ruthenium dioxide single-crystal surfaces. *Nat. Catal.* **3**, 516-525 (2020).

44. Velasco-Velez JJ, *et al.* Surface electron-hole rich species active in the electrocatalytic water oxidation. *J. Am. Chem. Soc.* **143**, 12524–12534 (2021).

3). Page 9, line 238-239, four references are added as follows.

The distinction in Tafel slopes implies probably modified OER pathway as discussed in following parts.^{45, 46, 47, 48}

45. Ping Y, Nielsen RJ, Goddard WA, 3rd. The reaction mechanism with free energy barriers at constant potentials for the oxygen evolution reaction at the IrO₂ (110) surface. *J. Am. Chem. Soc.* **139**, 149-155 (2017).

46. Liu C, *et al.* Oxygen evolution reaction over catalytic single-site Co in a well-defined brookite TiO₂ nanorod surface. *Nat. Catal.* **4**, 36-45 (2021).

47. Giordano L, *et al.* pH dependence of OER activity of oxides: Current and future perspectives. *Catal. Today* **262**, 2-10 (2016).

48. Tsuji E, Imanishi A, Fukui K-i, Nakato Y. Electrocatalytic activity of amorphous RuO₂ electrode for oxygen evolution in an aqueous solution. *Electrochim. Acta* **56**, 2009-2016 (2011).

4a). Page 9, line 255-258, three references are added as follows.

Sparse and uniformly dispersed particles appear on the surface without discernible depth but denser for 0.2Mo-PIO, manifesting quicker surface evolution because of increased Ir–O covalency and the consequent lattice oxygen activation.^{9, 13, 49}

9. Wan G, *et al.* Amorphization mechanism of SrIrO₃ electrocatalyst: How oxygen redox initiates ionic diffusion and structural reorganization. *Sci. Adv.* **7**, eabc7323 (2021).

13. Zhang N, Chai Y. Lattice oxygen redox chemistry in solid-state electrocatalysts for water oxidation. *Energy Environ. Sci.* **14**, 4647-4671 (2021).

49. Wu TZ, *et al.* Iron-facilitated dynamic active-site generation on spinel CoAl₂O₄ with self-termination of surface reconstruction for water oxidation. *Nat. Catal.* **2**, 763-772 (2019).

4b). Page 10, line 266-268, one reference is added as follows.

However, the obvious positive-shift of Mo 3d peaks manifest that the decrease in Ir oxidation state after 2 cycles is associated not only with lattice oxygen activation but with charge redistribution induced by Mo (Figure 5d).⁴²

42. Zhao F, *et al.* Increasing iridium oxide activity for the oxygen evolution reaction with hafnium modification. *J. Am. Chem. Soc.* **143**, 15616-15623 (2021).

4c). Page 11, line 280-281, three references are added as follows.

On the other hand, the consequent under-coordinated cation sites adjacent to oxygen vacancies on such metastable surface are more likely to dissolve under oxidation potentials.^{8, 9, 52}

8. Grimaud A, *et al.* Activation of surface oxygen sites on an iridium-based model catalyst for the oxygen evolution reaction. *Nat. Energy* **2**, 16189-16198 (2016).

9. Wan G, *et al.* Amorphization mechanism of SrIrO₃ electrocatalyst: How oxygen redox initiates ionic diffusion and structural reorganization. *Sci. Adv.* **7**, eabc7323 (2021).

52. Duan Y, *et al.* Anodic oxidation enabled cation leaching for promoting surface reconstruction in water oxidation. *Angew. Chem., Int. Ed.* **60**, 7418-7425 (2021).

4d). Page 11, line 289-292, two references are added as follows.

The positive-shifted O 1s peak with a higher content of -OH/O_{ads} species in PIO demonstrates charge compensation from further lattice oxygen loss when suffering from violent cation dissolution, consistent with the emergence of Ir^{III} species (Figure 5a, b).^{9, 36}

9. Wan G, *et al.* Amorphization mechanism of SrIrO₃ electrocatalyst: How oxygen redox initiates ionic diffusion and structural reorganization. *Sci. Adv.* **7**, eabc7323 (2021).

36. Pfeifer V, *et al.* In situ observation of reactive oxygen species forming on oxygen-evolving iridium surfaces. *Chem. Sci.* **8**, 2143-2149 (2017).

4e). Page 11, line 300-302, two references are added as follows.

The continuously decreased Ir oxidation state and the simultaneously positive-shifted O 1s peaks demonstrate further variation of surface oxygen species, implying an unstable kinetics of surface oxygen cycles (Figure 5a, b).^{53, 54}

53. Pfeifer V, *et al.* The electronic structure of iridium oxide electrodes active in water splitting. *Phys. Chem. Chem. Phys.* **18**, 2292-2296 (2016).

54. She L, Zhao G, Ma T, Chen J, Sun W, Pan H. On the durability of iridium-based electrocatalysts toward the oxygen evolution reaction under acid environment. *Adv. Funct. Mater.* **32**, 2108465 (2021).

Comment 7:

Please label the Raman peaks in Fig. 6e so it is easier to understand the structure evolution.

Response 7:

Thanks for the nice reminder from the reviewer. We have labelled relevant Raman peaks in Figure 6e.

Modification 7:

1). Figure 6e is revised as follows.

Figure 6. Role of reconstructed Ir–O_{bri}–Mo species in promoting deprotonation. (a) LSV curves for PIO-post and 0.2Mo-PIO-post in HClO₄ with varied pH. (b) OER current density at 1.53 V_{RHE} plotted in log scale as a function of pH, from which the proton reaction orders ($\rho^H = \partial \log(j) / \partial \text{pH}$) were calculated. (c) LSV curves for PIO-post and 0.2Mo-PIO-post measured in 0.1 M HClO₄ prepared in H₂O and D₂O. (d) KIE of PIO-post and 0.2Mo-PIO-post. j^H and j^D are referred to the current density measured in 0.1M HClO₄ prepared in H₂O and D₂O at the same overpotential, respectively. (e) *In situ* Raman spectra of PIO-post (red) and 0.2Mo-PIO-post (blue) in 0.1M HClO₄ at varied potentials.

Comment 8:

In figure 5b and 5c, the peak for Ir^{<V} is located at the lower energy compared to that for Ir^{III}, indicating that the valence is below +3. Please provide more details on this analysis.

Response 8:

We thank the reviewer for the comment. As supported by reported theory and experimental evidence, the main line peaks of Ir^{III} species appear at around 62.3 eV and 65.3 eV for Ir 4*f*_{7/2} and Ir 4*f*_{5/2}, respectively, which show positive shift of binding energies compared with those for Ir^{IV} species (around 61.7 eV and 64.7 eV for Ir 4*f*_{7/2} and Ir 4*f*_{5/2}, respectively). (*Surf. Interface Anal.* **48**, 261 (2016). *Phys. Chem. Chem. Phys.* **18**, 2292 (2016). *J. Electrochem. Soc.* **131** 72 (1984)). As a result, although the peak of Ir^{<V} species is located at the lower energy compared to that of Ir^{III} species, the average oxygen state is above +3. We have provided more detailed analysis in the revised manuscript.

Modification 8:

1). Page 10, line 263-265, one sentence is revised as follows

There is still Ir^V species remaining in PIO, which completely disappear but accompanying formation of abundant Ir^{III} species in 0.2Mo-PIO (Figure 5b, and 5c, Supporting Note 3).^{30, 31, 50}

30. Freakley SJ, Ruiz-Esquius J, Morgan DJ. The X-ray photoelectron spectra of Ir, IrO₂ and IrCl₃ revisited. *Surf. Interface Anal.* **49**, 794-799 (2017).

31. Pfeifer V, *et al.* The electronic structure of iridium and its oxides. *Surf. Interface Anal.* **48**, 261-273 (2016).

50. Kötz R, Neff H, Stucki S. Anodic iridium oxide films: XPS-studies of oxidation state changes and O₂-evolution. *J. Electrochem. Soc.* **131**, 72 (1984).

2). Supporting Information, Supporting Note 3 is added as follows.

Supporting Note 3

As supported by reported theory and experimental evidence, the main line peaks of Ir^{III} species appear at around 62.3 eV and 65.3 eV for Ir 4f_{7/2} and Ir 4f_{5/2}, respectively, which show positive shift of binding energies compared with those for Ir^{IV} species (around 61.7 eV and 64.7 eV for Ir 4f_{7/2} and Ir 4f_{5/2}, respectively). As a result, although the peak of Ir^{<V} species is located at the lower energy compared to that of Ir^{III} species, the average oxygen state is above +3. After 2 electrochemical cycles, the surface lattice oxygen oxidation (O₂ release) and subsequent occupation of oxygen vacancies with water molecules or hydroxyl contribute to the reduction of metal oxidation state. As shown in HRTEM and corresponding SAED images (Figure 4a, b right panel), sparse particles appear on the surface but with well-ordered bulk crystalline, indicating an ongoing surface reconstruction. So, the surface is not fully covered with hydroxide species IrO_x.

Comment 9:

In figure 2d, please double check the scale in the x axis.

Response 9:

We thank the reviewer for the critical comment. We have checked and corrected the x axis scale in Figure 2d.

Modification 9:

1). Figure 2d is revised as follows.

Figure 2. Structural characterizations of $x\text{Mo-PIO}$. (a) Powder XRD patterns. (b), (c) HRTEM images of PIO (b) and 0.2Mo-PIO (c). Insets: corresponding SAED patterns. The relevant interplanar spacing and dihedral angles are labeled. (d), (e) XPS spectra of Ir 4f (d) and O 1s (e) core-level with peak fitting results. (f) Atomic ratio comparison of fitting results and the binding energy changes for $-\text{OH}/\text{O}_{\text{ads}}$ (blue ball and blue triangle) and O_{L2} (red ball and red triangle). (g) EPR results.

Reviewer #3

The manuscript “Reconstructed Ir–O_{bri}–Mo species with strong Brønsted acidity for acidic water oxidation” by Chen et al describes the synthesis and investigation of $\text{Pr}_3\text{Ir}_{1-x}\text{Mo}_x\text{O}_7$ and the resulting Ir–O_{bri}–Mo active species by different methods, including DFT, XPS, Raman and electrochemical measurements. Currently, green hydrogen plays a crucial role in the global energy discussion, and with this comes also a special importance of iridium based OER catalysts. The here presented results show new insights and solution in how to improve the stability and activity of the iridium based catalysts. It is shown how the addition of new elements into Iridium oxide (namely Pr and Mo) can improve the performance of the catalyst, and also that the conversion during the OER has to be considered. The manuscript gives deep insights into the appearing processes and the properties of the resulting species. The graphs are presented in a very nice and clear manner and the discussion in the text is easy to understand and to follow. Overall, it is around the easiest to understand and most clear structured paper I have read within the last few months.

The combination of the theoretical calculations and the electrochemical testing, as well as the characterization of the catalyst before and after the catalyst testing, makes this a nice manuscript with high interest to the research community.

I therefore suggest this manuscript to be accepted for publications after addressing the following minor points:

Comment 1:

It is not totally clear to me how the authors choose Pr_3IrO_7 as a system to do DFT calculations on. Therefore, some sentences should be added at the beginning to make the line of thought of the authors clearer.

Response 1:

We thank the reviewer for the comment. We have added some sentences at the beginning to elucidate the line of the thought to choose Pr_3IrO_7 as a system to do DFT calculations on in the revised manuscript.

Modification 1:

1). Page 2, line 45-48, one sentence is revised as follows.

Weberite type Ln_3IrO_7 , featured with corner-linked IrO_6 octahedra lying along c -axis with shared apical $\text{O}_{(3)}$ and layered structure along a -axis, provides prerequisites to activate lattice oxygen redox under electrochemical conditions due to the large O 2p contribution around Fermi level (Figure 1a and 1d, Supporting Note 1).^{11, 14, 15, 16}

2). Page 3, line 82-84, one sentence is revised as follows.

Based on the flexible crystal structure and characteristic band structure of Pr_3IrO_7 , high-valent Mo is doped into Pr_3IrO_7 to accelerate surface reconstruction and obtain active reconstructed layers (Figure 1).

3). Supporting Information, two sentences are added in **Computational Methods**.

Pr_3IrO_7 crystal with $Cmcm$ space group was selected, and the optimal lattice parameter is $a = 7.55 \text{ \AA}$, $b = 11.07 \text{ \AA}$, $c = 7.63 \text{ \AA}$. For Mo-doped Pr_3IrO_7 , the Ir site is substituted with one Mo atom, and the corresponding lattice parameter have a little variation with $a=7.56 \text{ \AA}$, $b=11.08 \text{ \AA}$, $c=7.67 \text{ \AA}$.

Comment 2:

Likewise, a short discussion and outlook should be added at the end to put this research work into the wider perspective.

Response 2:

Thanks for the comment from the reviewer. We have revised the part **3. Conclusions** in the revised manuscript.

Modification 2:

1). Page 17, line 443-452, one paragraph is revised as follows.

In summary, using $\text{Pr}_3\text{Ir}_{1-x}\text{Mo}_x\text{O}_7$ as model, we have developed a facile electrochemical reconstruction strategy to construct highly active and self-terminated Ir–O_{br}–Mo species for water oxidation in acidic electrolyte. The presence of high-valence Mo accelerates surface reconstruction due to the optimized Ir–O covalency and more prone dissolution of Pr. Meanwhile, excessive loss of lattice oxygen is effectively avoided benefitting from Mo-buffered charge compensation. Significantly, highly active Ir–O_{br}–Mo species as strong Brønsted acid in surface reconstruction layers facilitate deprotonation of oxo intermediates following BOAD pathway, resulting in an overall activity improvement. This work proposes a facile strategy for constructing strong Brønsted acid sites in IrO_x through directional surface reconstruction of iridates, demonstrating the perspective of targeted electrocatalyst fabrication under in-situ realistic reaction conditions.

REVIEWERS' COMMENTS

Reviewer #1 (Remarks to the Author):

The authors have addressed my previous comments in a satisfactory fashion. The computational results are now physically more sound and should be reproducible. The language (also of the added sections) still needs some polishing, which can hopefully occur during copy-editing.

Reviewer #2 (Remarks to the Author):

The manuscript has been significantly improved after the revision.

Only there is typo in the caption of Figure S14.